# BoxE: A Box Embedding Model for Knowledge Base Completion

**Ralph Abboud, İsmail İlkan Ceylan, Thomas Lukasiewicz, Tommaso Salvatori**
Department of Computer Science
University of Oxford, UK
`{firstame.lastname}@cs.ox.ac.uk`

## Abstract

Knowledge base completion (KBC) aims to automatically infer missing facts by exploiting information already present in a knowledge base (KB). A promising approach for KBC is to embed knowledge into latent spaces and make predictions from learned embeddings. However, existing embedding models are subject to at least one of the following limitations: (1) theoretical *inexpressivity*, (2) lack of support for prominent *inference patterns* (e.g., hierarchies), (3) lack of support for KBC over *higher-arity* relations, and (4) lack of support for incorporating *logical rules*. Here, we propose a *spatio-translational* embedding model, called *BoxE*, that simultaneously addresses all these limitations. BoxE embeds entities as *points*, and relations as a set of *hyper-rectangles* (or *boxes*), which spatially characterize basic logical properties. This seemingly simple abstraction yields a fully expressive model offering a natural encoding for many desired logical properties. BoxE can both *capture* and *inject* rules from rich classes of rule languages, going well beyond individual inference patterns. By design, BoxE naturally applies to higher-arity KBs. We conduct a detailed experimental analysis, and show that BoxE achieves state-of-the-art performance, both on benchmark knowledge graphs and on more general KBs, and we empirically show the power of integrating logical rules.

## 1 Introduction

Knowledge bases (KBs) are fundamental means for representing, storing, and processing information, and are widely used to enhance the *reasoning* and *learning* capabilities of modern information systems. KBs can be viewed as a collection of facts of the form $r(e_1, \ldots, e_n)$, which represent a relation $r$ between the entities $e_1, \ldots, e_n$, and knowledge graphs (KGs) as a special case, where all the relations are binary (i.e., composed of two entities). KBs such as YAGO [24], NELL [26], Knowledge Vault [9], and Freebase [2] contain millions of facts, and are increasingly important in academia and industry, for applications such as question answering [3], recommender systems [39], information retrieval [44], and natural language processing [45].

KBs are, however, highly *incomplete*, which makes their downstream use more challenging. For instance, 71% of individuals in Freebase lack a connection to a place of birth [42]. *Knowledge base completion (KBC)*, aiming at automatically inferring missing facts in a KB by exploiting the already present information, has thus become a focal point of research. One prominent approach for KBC is to learn *embeddings* for entities and relations in a latent space such that these embeddings, once learned from known facts, can be used to *score* the plausibility of unknown facts.

Currently, the main embedding approaches for KBC are translational models [4, 34], which score facts based on distances in the embedding space, bilinear models [36, 46, 1], which learn embeddings that factorize the truth tensor of a knowledge base, and neural models [7, 31, 27], which score facts using dedicated neural architectures. Each of these models suffer from limitations, most of

which are well-known. Translational models, for instance, are theoretically inexpressive, i.e., cannot provably fit an arbitrary KG. Furthermore, none of these models can capture simple sets of *logical rules*: even capturing a simple relational hierarchy goes beyond the current capabilities of most existing models [13]. This also makes it difficult to inject background knowledge (i.e., schematic knowledge), in the form of logical rules, into the model to improve KBC performance. Additionally, existing KBC models are primarily designed for KGs, and thus do not naturally extend to KBs with *higher-arity* relations, involving 3 or more entities, e.g., DegreeFrom(Turing, PhD, Princeton) [10], which hinders their applicability. Higher-arity relations are prevalent in modern KBs such as Freebase [41], and cannot always be reduced to a KG without loss of information [10]. Despite the rich landscape for KBC, no existing model currently offers a solution to all these limitations.

In this paper, we address these problems by encoding relations as explicit regions in the embedding space, where logical properties such as relation subsumption and disjointness can naturally be analyzed and inferred. Specifically, we present *BoxE*, a *spatio-translational* box embedding model, which models relations as sets of $d-$dimensional boxes (corresponding to classes), and entities as $d-$dimensional points. Facts are scored based on the positions of entity embeddings with respect to relation boxes. Our contributions can be summarized as follows:

– We introduce BoxE and show that this model achieves state-of-the-art performance on both *knowledge graph completion* and *knowledge base completion* tasks across multiple datasets.

– We show that BoxE is fully expressive, a first for translation-based models, to our knowledge.

– We comprehensively analyze the inductive capacity of BoxE in terms of generalized inference patterns and rule languages, and show that BoxE can capture a rich rule language.

– We prove that BoxE additionally supports *injecting* a rich language of logical rules, and empirically show on a subset of NELL [26], that this can significantly improve KBC performance.

All proofs for theorems, as well as experimental details, can be found as an appendix in the long version of this paper.

## 2   Knowledge Base Completion: Problem, Properties, and Evaluation

In this section, we define knowledge bases and the problem of knowledge base completion (KBC). We also give an overview of standard approaches for evaluating KBC models.

Consider a *relational vocabulary*, which consists of a finite set $\mathbf{E}$ of *entities* and a finite set $\mathbf{R}$ of *relations*. A *fact* (also called *atom*) is of the form $r(e_1, \ldots, e_n)$, where $r \in \mathbf{R}$ is an $n$-ary relation, and $e_i \in \mathbf{E}$ are entities. A *knowledge base (KB)* is a finite set of facts, and a *knowledge graph (KG)* is a KB with only binary relations. In KGs, facts are also known as *triples*, and are of the form $r(e_h, e_t)$, with a *head* entity $e_h$ and a *tail* entity $e_t$. *Knowledge base completion (KBC)* (resp., knowledge graph completion (KGC)) is the task of accurately predicting new facts from existing facts in a KB (resp., KG). KBC models are analyzed by means of (i) an *experimental evaluation* on existing benchmarks, (ii) their model *expressiveness*, and (iii) the set of *inference patterns* that they can capture.

**Experimental evaluation.** To evaluate KBC models empirically, *true facts* from the test set of a KB and *corrupted facts*, generated from the test set, are used. A corrupted fact is obtained by replacing one of the entities in a fact from the KB with a new entity: given a fact $r(e_1, \ldots, e_i, \ldots, e_n)$ from the KB, a corrupted fact is a fact $r(e_1, \ldots, e_i', \ldots, e_n)$ that does *not* occur in the training, validation, or test set. KBC models define a *scoring function* over facts, and are optimized to score true facts higher than corrupted facts. KBC performance is evaluated using metrics [4] such as mean rank (MR), the average rank of facts against their corrupted counterparts, mean reciprocal rank (MRR), their average inverse rank (i.e., 1/rank), and Hits@K, the proportion of facts with rank at most K.

**Expressiveness.** A KBC model $\mathcal{M}$ is *fully expressive* if, for any given disjoint sets of *true* and *false* facts, there exists a parameter configuration for $\mathcal{M}$ such that $\mathcal{M}$ accurately classifies all the given facts. Intuitively, a fully expressive model can capture any knowledge base configuration, but this does not necessarily correlate with inductive capacity: fully expressive models can merely memorize training data and generalize poorly. Conversely, a model that is not fully expressive can fail to fit its training set properly, and thus can underfit. Hence, it is important to develop models that are jointly fully expressive and capture prominent and common inference patterns.

**Inference patterns.** Inference patterns are a common means to formally analyze the generalization ability of KBC systems. Briefly, an *inference pattern* is a specification of a logical property that may exist in a KB, which, if learned, enables further principled inferences from existing KB facts. One well-known example inference pattern is *symmetry*, which specifies that when a fact $r(e_1, e_2)$ holds, then $r(e_2, e_1)$ also holds. If a model learns a symmetry pattern for $r$, then it can automatically predict facts in the symmetric closure of $r$, thus providing a strong inductive bias. We present some prominent inference patterns in detail in Section 5, and also in Table 1. Intuitively, inference patterns captured by a model serve as an indication of its *inductive capacity*.

## 3   Related Work

In this section, we give an overview of closely related embedding methods for KBC/KGC and existing region-based embedding models. We exclude neural models [7, 32, 27], as these models are challenging to analyze, both from an expressiveness and inductive capacity perspective.

**Translational models.** Translational models represent entities as points in a high-dimensional vector space and relations as translations in this space. The seminal translational model is TransE [4], where a relation $r$, modeled by a vector $\boldsymbol{r}$, holds between $e_1$ and $e_2$ iff $\boldsymbol{e_1} + \boldsymbol{r} = \boldsymbol{e_2}$. However, TransE is not fully expressive, cannot capture *one-to-many*, *many-to-one*, *many-to-many*, and symmetric relations, and can only handle binary facts. This motivated extensions [40, 21, 16, 11], which each address some, but not all, these limitations. Beyond translations, RotatE [34] uses rotations to model relations, and thus can model symmetric relations with rotations of angle $\theta = \pm\pi$, but is otherwise as limited as TransE. Translational models are interpretable and can capture various inference patterns, but no known translational model is fully expressive.

**Bilinear models.** Bilinear models capture relations as a bilinear product between entity and relation embeddings. RESCAL [28] represents a relation $r$ as a full-rank $d \times d$ matrix $M$, and entities as $d$-dimensional vectors $\boldsymbol{e}$. DistMult [46] simplifies RESCAL by making $M$ diagonal, but cannot capture non-symmetric relations. ComplEx [36] defines a diagonal $M$ with complex numbers to capture anti-symmetry. SimplE [17] and TuckER [1] build on canonical polyadic (CP) [14] and Tucker decomposition [37], respectively. TuckER subsumes RESCAL, its adaptations, and SimplE [1]. Generally, all bilinear models except DistMult are fully expressive, but they are less interpretable compared to translational models.

**Higher-arity KBC.** KBs can encode knowledge that cannot be encoded in a KG [10]. Hence, models such as HSimplE [10], m-TransH [41], m-DistMult, and m-CP [10] are proposed as generalizations of SimplE, TransH [40], DistMult, and CP, respectively. HypE [10] tackles higher-arity KBC through convolutions. Generalizations to TuckER, namely, m-TuckER and GETD [22], are also proposed, but these do not apply to KBs with different-arity relations. For most existing KGC models, there are conceptual and practical challenges (e.g., scalability) against generalizing them to KBC.

**Region-based models.** Region-based models explicitly define regions in the embedding space where an output property (e.g., membership to a class) holds. For instance, bounded axis-aligned hyper-rectangles (boxes) [38, 33, 20] are used for entity classification to define class regions and hierarchies, in which entity point embeddings appear. As boxes naturally represent sets of objects, they are also used to represent answer sets in the Query2Box query answering system [15]. Query2Box can be applied to KBC but reduces to a translational model with a box correctness region for tail entities. Furthermore, entity classification approaches cannot be scalably generalized to KBC, as this would involve introducing an embedding per entity tuple.

## 4   Box Embeddings for Knowledge Base Completion

In this section, we introduce an embedding model for KBC, called *BoxE*, that encodes relations as axis-aligned *hyper-rectangles* (or boxes) and entities as *points* in the $d$-dimensional Euclidian space.

**Representation.** In BoxE, every entity $e_i \in \mathbf{E}$ is represented by two vectors $\boldsymbol{e_i}, \boldsymbol{b_i} \in \mathbb{R}^d$, where $\boldsymbol{e_i}$ defines the *base position* of the entity, and $\boldsymbol{b_i}$ defines its *translational bump*, which translates all the entities co-occuring in a fact with $e_i$, from their base positions to their final embeddings by "bumping"

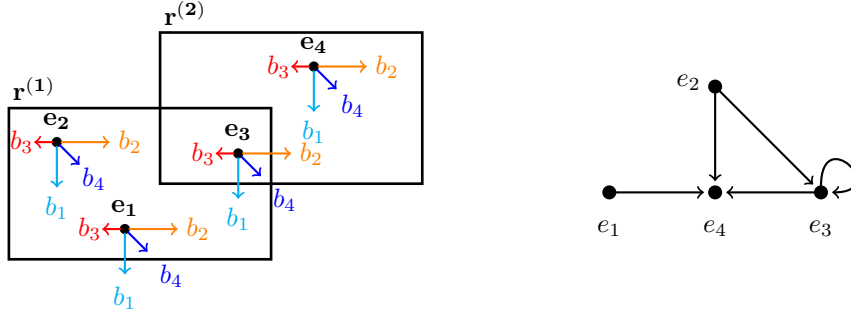

Figure 1: A sample BoxE model is shown on the left for $d = 2$. The binary relation $r$ is encoded via the box embeddings $\mathbf{r^{(1)}}$ and $\mathbf{r^{(2)}}$. Every entity $e_i$ has an embedding $\mathbf{e_i}$, and defines a bump on other entities, as shown with distinct colors. This model induces the KG on $r$, shown on the right.

them. The *final embedding* of an entity $e_i$ relative to a fact $r(e_1, \ldots, e_n)$ is hence given by:

$$\boldsymbol{e_i}^{r(e_1,\ldots,e_n)} = (\boldsymbol{e_i} - \boldsymbol{b_i}) + \sum_{1 \leq j \leq n} \boldsymbol{b_j}. \tag{1}$$

Essentially, the entity representation is dynamic, as every entity can have a potentially different final embedding relative to a different fact. The main idea is that every entity translates the base positions of other entities co-appearing in a fact, that is, for a fact $r(e_1, e_2)$, $\boldsymbol{b_1}$ and $\boldsymbol{b_2}$ translate $\boldsymbol{e_2}$ and $\boldsymbol{e_1}$ respectively, to compute their final embeddings.

In BoxE, every relation $r$ is represented by $n$ hyper-rectangles, i.e., boxes, $\boldsymbol{r^{(1)}}, \ldots, \boldsymbol{r^{(n)}} \in \mathbb{R}^d$, where $n$ is the arity of $r$. Intuitively, this representation defines $n$ *regions* in $\mathbb{R}^d$, one per arity position, such that a fact $r(e_1, ..., e_n)$ holds when the final embeddings of $e_1, ..., e_n$ each appear in their corresponding position box, creating a *class* abstraction for the sets of all entities appearing at every arity position. For the special case of unary relations (i.e., classes), the definition given in Eq. 1 implies no translational bumps, and thus the base position of an entity is its final embedding.

**Example 4.1.** Consider an example over a single binary relation $r$ and the entities $e_1, e_2, e_3, e_4$. A BoxE model is given on the left in Figure 1, for $d = 2$, where every entity is represented as a point, and the binary relation $r$ is represented with two boxes $\boldsymbol{r^{(1)}}$ and $\boldsymbol{r^{(2)}}$. Every entity is translated by the bump vectors of all other entities. For example, $r(e_1, e_4)$ is a true fact in the model (e.g., to be ranked high), since (i) $\boldsymbol{e_1}^{r(e_1,e_4)} = (\boldsymbol{e_1} + \boldsymbol{b_4})$ is a point in $\boldsymbol{r^{(1)}}$ ($e_1$ appears in the head box), and (ii) $\boldsymbol{e_4}^{r(e_1,e_4)} = (\boldsymbol{e_4} + \boldsymbol{b_1})$ is a point in $\boldsymbol{r^{(2)}}$ ($e_4$ appears in the tail box). Similarly, $r(e_3, e_3)$ is a true fact in the model, as $\boldsymbol{e_3}^{r(e_3,e_3)} = (\boldsymbol{e_3} + \boldsymbol{b_3})$, which is a point in $\boldsymbol{r^{(1)}}$ and $\boldsymbol{r^{(2)}}$, i.e., the entity is reflexive in $r$. The model encodes all (and only) the facts from the KG, shown on the right in Figure 1. ∎

Translational bumps are very powerful, as they allow us to model complex interactions across entities in an effective manner. Observe that for the sample KG, there are $4^2$ potential facts that can hold, and therefore $2^{16}$ possible configurations. Nonetheless, they can all be compactly captured by choosing appropriate translational bumps to force entity embeddings in or out of the respective relation boxes as needed. Indeed, we later formally show that such a configuration can always be found for any KB, given sufficiently many dimensions, proving full expressiveness of the model.

**Scoring function.** In the above example, we identified facts that ideally need to be ranked higher by our scoring function, to reflect the model properties adequately. To this end, we first define a distance function for evaluating entity positions relative to the box positions. The idea is to define a function that grows slowly if a point is in the box (relative to the center of the box), but grows rapidly if the point is outside of the box, so as to drive points more effectively into their target boxes and ensure they are minimally changed, and can remain there once inside.

Formally, let us denote by $\boldsymbol{l^{(i)}}, \boldsymbol{u^{(i)}} \in \mathbb{R}^d$ the *lower* and *upper* boundaries of a box $\boldsymbol{r^{(i)}}$, respectively, by $\boldsymbol{c^{(i)}} = (\boldsymbol{l^{(i)}} + \boldsymbol{u^{(i)}})/2$ its center, and by $\boldsymbol{w^{(i)}} = \boldsymbol{u^{(i)}} - \boldsymbol{l^{(i)}} + 1$ its width incremented by 1. We say that a point $\boldsymbol{e_i}$ is inside a box $\boldsymbol{r^{(i)}}$, denoted $\boldsymbol{e_i} \in \boldsymbol{r^{(i)}}$, if $\boldsymbol{l^{(i)}} \leq \boldsymbol{e_i} \leq \boldsymbol{u^{(i)}}$. Furthermore, we denote

the element-wise multiplication, division, and inversion operations by $\circ, \oslash$ and $^{\circ-1}$ respectively. Then, the *distance function* for the given entity embeddings relative to a given target box is defined piece-wise over two cases, as follows:

$$\mathsf{dist}(\boldsymbol{e}_i^{r(e_1,...,e_n)}, \boldsymbol{r}^{(i)}) = \begin{cases} |\, \boldsymbol{e}_i^{r(e_1,...,e_n)} - \boldsymbol{c}^{(i)}\,| \oslash \boldsymbol{w}^{(i)} & \text{if } \boldsymbol{e}_i \in r^{(i)}, \\ |\, \boldsymbol{e}_i^{r(e_1,...,e_n)} - \boldsymbol{c}^{(i)}\,| \circ \boldsymbol{w}^{(i)} - \kappa & \text{otherwise,} \end{cases}$$

where $\kappa = 0.5 \circ (\boldsymbol{w}^{(i)} - 1) \circ (\boldsymbol{w}^{(i)} - \boldsymbol{w}^{(i)^{\circ-1}})$, is a width-dependent factor.

In both cases, dist factors in the size of the target box in its computation. In the first case, where the point is in its target box, distance inversely correlates with box size, to maintain low distance inside large boxes and provide a gradient to keep points inside. In the second case, box size linearly correlates with distance, to penalize points outside larger boxes more severely. Finally, $\kappa$ is subtracted to preserve function continuity.

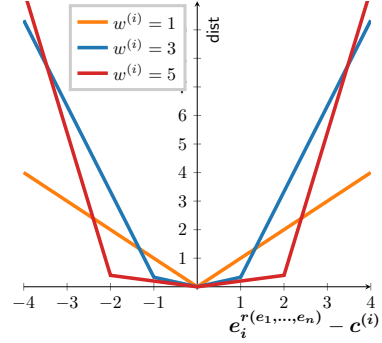

Plots for dist for one-dimensional $\boldsymbol{w}^{(i)}$ are shown in Figure 2. Observe that, when $\boldsymbol{w}^{(i)} = 1$, $\boldsymbol{r}^{(i)}$ is point-shaped, and dist reduces to standard $\mathrm{L}^1$ distance. Conversely, as $\boldsymbol{w}^{(i)}$ increases, dist gives lower values (and gradients) to the region inside the box, and severely punishes points outside. This function thus achieves three objectives. First, it treats points inside the box preferentially to points outside the box, unlike standard distance, which is agnostic to boxes. Second, it ensures that outside points receive high gradient through which they can

Figure 2: The dist function for width $\boldsymbol{w}^{(i)} = 1, 3, 5$.

more easily reach their target box, or escape it for negative samples. Third, it gives weight to the size of a box in distance computation, to yield a more comprehensive scoring mechanism.

Finally, we define the scoring function as the sum of the L-$x$ norms of dist across all $n$ entities and relation boxes, i.e.:

$$\mathsf{score}(r(e_1, ..., e_n)) = \sum_{i=1}^{n} \left\| \mathsf{dist}(\boldsymbol{e}_i^{r(e_1,...,e_n)}, \boldsymbol{r}^{(i)}) \right\|_x.$$

# 5 Model Properties

We analyze the representation power and inductive capacity of BoxE and show that BoxE is fully expressive, and can capture a rich language combining multiple inference patterns. We additionally show that BoxE can lucidly *incorporate* a given set of logical rules from a sublanguage of this language, i.e., rule injection. Finally, we analyze the complexity of BoxE in the appendix, and prove that it runs in time $O(nd)$ and space $O((|\mathbf{E}| + n|\mathbf{R}|)d)$, where $n$ is the maximal relation arity.

## 5.1 Full expressiveness

We prove that BoxE is fully expressive with $d = |\mathbf{E}|^{n-1}|\mathbf{R}|$ dimensions. For KGs, this result implies $d = |\mathbf{E}||\mathbf{R}|$, so BoxE is fully expressive over KGs with dimensionality *linear* in $|\mathbf{E}|$. The proof uses translational bumps to make an arbitrary true fact $F$ false, while preserving the correctness of other facts. This result requires a careful technical construction, which (i) pushes a single entity representation within $F$ outside its corresponding relation box at a specific dimension, and (ii) modifies all other model embeddings to prevent a change in the truth value of any other fact.

**Theorem 5.1.** *BoxE is a fully expressive model with the embedding dimensionality $d$ of entities, bumps, and relations set to $d = |\mathbf{E}|^{n-1}|\mathbf{R}|$, where $n > 1$ is the maximal arity of the relations in $\mathbf{R}$.*

We note that this result makes BoxE the first translation-based model that is fully expressive.

## 5.2 Inference patterns and generalizations

We study the inductive capacity of BoxE in terms of common inference patterns appearing in the KGC literature, and compare it with earlier models. A comparison of BoxE against these models with respect to capturing prominent inference patterns is shown in Table 1.

Table 1: Inference patterns/generalized inference patterns captured by selected KBC models. TuckER coincides with ComplEx, so is omitted from the table.

| Inference pattern | BoxE | TransE | RotatE | DistMult | ComplEx |
|---|---|---|---|---|---|
| Symmetry: $r_1(x,y) \Rightarrow r_1(y,x)$ | ✓/✓ | ✗/✗ | ✓/✓ | ✓/✓ | ✓/✓ |
| Anti-symmetry: $r_1(x,y) \Rightarrow \neg r_1(y,x)$ | ✓/✓ | ✓/✓ | ✓/✓ | ✗/✗ | ✓/✓ |
| Inversion: $r_1(x,y) \Leftrightarrow r_2(y,x)$ | ✓/✓ | ✓/✗ | ✓/✓ | ✗/✗ | ✓/✓ |
| Composition: $r_1(x,y) \wedge r_2(y,z) \Rightarrow r_3(x,z)$ | ✗/✗ | ✓/✗ | ✓/✗ | ✗/✗ | ✗/✗ |
| Hierarchy: $r_1(x,y) \Rightarrow r_2(x,y)$ | ✓/✓ | ✗/✗ | ✗/✗ | ✓/✗ | ✓/✗ |
| Intersection: $r_1(x,y) \wedge r_2(x,y) \Rightarrow r_3(x,y)$ | ✓/✓ | ✓/✗ | ✓/✗ | ✗/✗ | ✗/✗ |
| Mutual exclusion: $r_1(x,y) \wedge r_2(x,y) \Rightarrow \bot$ | ✓/✓ | ✓/✓ | ✓/✓ | ✓/✗ | ✓/✗ |

A model *captures* an inference pattern if it admits a set of parameters *exactly* and *exclusively* satisfying the pattern. This is the standard definition of an inference pattern in the literature [34]. For example, TransE can capture composition [4, 34], but cannot capture hierarchy, as for TransE, $r_1(x,y) \Rightarrow r_2(x,y)$ holds only if $r_1 = r_2$, and thus $r_2(x,y) \Rightarrow r_1(x,y)$, leading to loss of generality. However, this definition only addresses *single* applications of an inference pattern, which raises the question: can KBC models capture *multiple, distinct* instances of the *same* inference pattern jointly?

Capturing multiple inference patterns jointly is significantly more challenging. Indeed, TransE can capture $r_1(x,y) \wedge r_2(y,z) \Rightarrow r_3(x,z)$ and $r_1(x,y) \wedge r_4(y,z) \Rightarrow r_3(x,z)$ independently, but jointly capturing these compositions incorrectly forces $r_2 \sim r_4$. Similarly, bilinear models can capture the hierarchy rules $r_1(x,y) \Rightarrow r_3(x,y)$ and $r_2(x,y) \Rightarrow r_3(x,y)$ separately, but jointly capturing them incorrectly imposes either $r_1(x,y) \Rightarrow r_2(x,y)$ or $r_2(x,y) \Rightarrow r_1(x,y)$ [13]. These examples are clearly not edge cases, and highlight severe limitations in how the inductive capacity of KBC models is analyzed. Therefore, we propose and study *generalized inference patterns*.

**Definition 5.1.** *A rule is in one of the forms given in Table 1, where $r_1 \neq r_2 \neq r_3 \in \mathbf{R}$. To distinguish between types of rules, we write $\sigma$ rule, where $\sigma \in \{symmetry, \dots, mutual\ exclusion\}$. A generalized $\sigma$ pattern is a finite set of $\sigma$ rules over $\mathbf{R}$.*

As before, a model *captures* a generalized inference pattern if the model admits a set of parameters, exactly and exclusively satisfying the generalized pattern. Our results for BoxE and all relevant models are summarized in Table 1, and proven in the following theorem.

**Theorem 5.2.** *All the results given in Table 1 for BoxE and other models hold.*

Intuitively, BoxE captures all these generalized inference patterns through box configurations. For instance, BoxE captures (generalized) symmetry by setting the 2 boxes for a relation $r$ to be equal, and captures (generalized) inverse relations $r_1$ and $r_2$ by setting $r_1^{(1)} = r_2^{(2)}$ and $r_1^{(2)} = r_2^{(1)}$. Hierarchies are captured through box subsumption, i.e., $r_1^{(1)}$ and $r_1^{(2)}$ contained in $r_2^{(1)}$ and $r_2^{(2)}$ respectively, and this extends to intersection in the usual sense. Finally, anti-symmetry and mutual exclusion, are captured through disjointness between relation boxes.

Generalized inference patterns are necessary to establish a more complete understanding of model inductive capacity, and, in this respect, our results show that BoxE goes well beyond any other model. However, generalized patterns are not sufficient. Indeed, different types of inference rules can appear *jointly* in practical applications, so KBC models must be able to jointly capture them. This is not the case for existing models. For instance, RotatE can capture composition and generalized symmetry, but to capture a single composition rule such as $\mathsf{cousins}(x,y) \wedge \mathsf{hasChild}(y,z) \Rightarrow \mathsf{relatives}(x,z)$, where relatives and cousins are *symmetric* relations, the model forces hasChild to be symmetric as well, i.e., $\mathsf{hasChild}(x,y) \Rightarrow \mathsf{hasChild}(y,x)$, which is clearly absurd. Therefore, we also evaluate model inductive capacity relative to more general *rule languages* [13]. We define a rule language as the *union* of different types of rules. Thus, generalized inference patterns are trivial rule languages allowing only one type of rule. BoxE can capture rules from a rich language, as stated next.

**Theorem 5.3.** *Let $\mathcal{L}$ be the rule language that is the union of inverse, symmetry, hierarchy, intersection, mutual exclusion, and anti-symmetry rules. BoxE can capture any finite set of consistent rules from the rule language $\mathcal{L}$.*

This result captures generalized inference patterns for BoxE as a special case. Such a result is implausible for other KBC models, given their limitations in capturing generalized inference patterns, and we are unaware of any analogous result in KBC. The only related result is for ontology embeddings, and for quasi-chained rules [13], but this result merely offers region structures enabling capturing a set of rules, without providing any viable model or means of doing so.

The strong inductive capacity of BoxE is advantageous from an interpretability perspective, as all the rules that BoxE can jointly capture can be simply "read" from the corresponding box configuration. Indeed, BoxE embeddings allow for rich rule extraction, and enable an informed understanding of what the model learns, and how it reaches its scores. This is a very useful consequence of inductive capacity, as better rule capturing directly translates into superior model interpretability. Finally, BoxE can seamlessly and naturally represent *entity type* information, e.g., $\mathsf{country}(\mathsf{UK})$ by modeling types as *unary* relations. In this setting, translational bumps are not applicable, and inference patterns deducible from classic box configurations can additionally be captured and extracted. By contrast, standard models require dedicated modifications to their parameters and scoring function [43, 5, 23] to incorporate type information. This therefore further highlights the strong inductive capacity of BoxE, and its position as a unifying model for multi-arity knowledge base completion.

### 5.3 Rule injection

We now pose a complementary question to capturing inference patterns: can a KBC model be injected with a *given* set of rules such that it provably enforces them, improving its prediction performance? Formally, we say that a rule $\phi \Rightarrow \psi$ (resp., $\psi \Leftrightarrow \phi$) can be *injected* to a model, if the model can be configured to force $\psi$ to hold whenever $\phi$ holds (resp., $\phi$ holds whenever $\psi$ holds and vice versa).

There is a subtle difference between *capturing* and *injecting* an inference pattern. Indeed, rules with negation, such a mutual exclusion, can be easily captured with any disjointness between $r_1$ and $r_2$, but enforcing such a rule leads to non-determinism. To illustrate, $r_1$ and $r_2$ can be disjoint between their (i) head boxes, or (ii) tail boxes, or (iii) both, and at any combination of dimensions. This non-determinism only becomes more intricate as interactions across different rules are considered. We show that the positive fragment of the rule language that can be captured by BoxE, can be injected.

**Theorem 5.4.** *Let $\mathcal{L}^+$ be the rule language that is the union of inverse, symmetry, hierarchy, and intersection rules. BoxE can be injected with any finite set of rules from the rule language $\mathcal{L}^+$.*

Existing KGC rule injection methods (i) use rule-based training loss to inject rules [6, 29], potentially leveraging fuzzy logic [12] and adversarial training [25], but cannot provably enforce rules, or (ii) constrain embeddings explicitly [8, 29], but only enforce very limited rules (e.g., inversion, linear implication). Indeed, most popular standard KGC methods fail to capture simple sets of rules [13]. BoxE is a powerful model for rule injection in that it can explicitly and provably enforce such rules and incorporate a strong bias by appropriately constraining the learning space. Our study is related to the broader goal of making gradient-based optimization and learning compatible with reasoning [19].

## 6 Experimental Evaluation

In this section, we evaluate BoxE on a variety of tasks, namely, KGC, higher-arity KBC, and rule injection, and report state-of-the-art results, empirically confirming the theoretical strengths of BoxE.

### 6.1 Knowledge graph completion

In this experiment, we run BoxE on the KGC benchmarks FB15k-237, WN18RR, and YAGO3-10, and compare it with translational models TransE [4] and RotatE [34], both with uniform and self-adversarial negative sampling [34], and with bilinear models DistMult [46], ComplEx [36], and TuckER [1]. We train BoxE for up to 1000 epochs, with validation checkpoints every 100 epochs and the checkpoint with highest MRR used for testing. We report the best published results on every dataset for all models, and, when unavailable, report our best computed results in italic. All results are for models with $d \leq 1000$, to maintain comparison fairness [1]. We therefore exclude results by ComplEx [18] and DistMult [30] using $d \geq 2000$. The best results by category are presented in bold, and the best results overall are highlighted by a surrounding rectangle. "(u)" indicates uniform negative sampling, and "(a)" denotes self-adversarial sampling. Further details about experimental setup, as well as hyperparameter choices and dataset properties, can be found in the appendix.

Table 2: KGC results (MR, MRR, Hits@10) for BoxE and competing approaches on FB15k-237, WN18RR, and YAGO3-10. Other approach results are best published, with sources cited per model.

| Model | FB15k-237 | | | WN18RR | | | YAGO3-10 | | |
|---|---|---|---|---|---|---|---|---|---|
| | MR | MRR | H@10 | MR | MRR | H@10 | MR | MRR | H@10 |
| TransE(u) [30] | - | .313 | .497 | - | .228 | .520 | - | - | - |
| RotatE(u) [34] | 185 | .297 | .480 | *3254* | **.470** | **.564** | *1116* | *.459* | *.651* |
| BoxE(u) | **172** | **.318** | **.514** | 3117 | .442 | .523 | 1164 | **.567** | **.699** |
| TransE(a) [34] | 170 | .332 | .531 | 3390 | .223 | .529 | - | - | - |
| RotatE(a) [34] | 177 | **.338** | .533 | 3340 | **.476** | **.571** | 1767 | .495 | .670 |
| BoxE(a) | 163 | .337 | **.538** | 3207 | .451 | .541 | 1022 | .560 | **.691** |
| DistMult [30, 46] | - | .343 | .531 | - | .452 | .531 | 5926 | .34 | .54 |
| ComplEx [30, 46] | - | .348 | .536 | - | **.475** | **.547** | 6351 | .36 | .55 |
| TuckER [1] | - | .358 | .544 | - | .470 | .526 | *4423* | *.529* | *.670* |

**Results.** For every dataset and model, MR, MRR, and Hits@10 are reported in Table 2. On FB15k-237, BoxE performs best among translational models, and is competitive with TuckER, especially in Hits@10. Furthermore, BoxE is comfortably state-of-the-art on YAGO3-10, significantly surpassing RotatE and TuckER. This result is especially encouraging considering that YAGO3-10 is the largest of all three datasets, and involves a challenging combination of inference patterns, and many fact appearances per entity. On YAGO3-10, we also observe that BoxE successfully learns symmetric relations, and learns box sizes correlating strongly with relational properties (cf. Appendix). Strong BoxE performance on FB15k-237, which contains several composition patterns, suggests that BoxE can perform well with compositions, despite not capturing them explicitly as an inference pattern.

On WN18RR, BoxE performs well in terms of MR, but is less competitive with RotatE in MRR. We investigated WN18RR more deeply, and identified two main factors for this. First, WN18RR primarily consists of hierarchical knowledge, which is logically flattened into deep tree-shaped compositions, such as hypernym(spoon, utensil). Second, symmetry is prevalent in WN18RR, e.g., derivationally_related_form accounts for 29,715 (~34.5%) of WN18RR facts, which, combined with compositions, also helps RotatE. Indeed, in RotatE, the composition of two symmetric relations is (incorrectly) symmetric, but this is useful for WN18RR, where 4 of the the 11 relations are symmetric. That is, the modelling limitations of RotatE become an advantage given the setup of WN18RR, and enable it to achieve state-of-the-art performance on this dataset.

Overall, BoxE is competitive on all benchmarks , and is state of the art on YAGO3-10. Hence, it is a strong model for KGC on large, real-world KGs. We also evaluated the robustness of BoxE relative to dimensionality on YAGO3-10, and analyzed the resulting box configuration on this dataset from an interpretability perspective. These additional experiments can be found in the appendix.

## 6.2 Higher-arity knowledge base completion

In this experiment, we evaluate BoxE on datasets with *higher arity*, namely the publicly available JF-17K and FB-AUTO. These datasets contain facts with arities up to 6 and 5, respectively, and include facts with *different arities*, i.e., 2, 3, 4, and 5. We compare BoxE with the best-known reported results over the same datasets [10]. For this experiment, we set $d = 200$, for fairness with other models, and perform hyperparameter tuning analogously to Section 6.1.

**Results.** MRR and Hits@10 for all evaluated models are given in Table 3. On both datasets, BoxE achieves state-of-the-art performance.

Table 3: KBC results on JF17K and FB-AUTO.

| Model | JF17K | | FB-AUTO | |
|---|---|---|---|---|
| | MRR | H@10 | MRR | H@10 |
| m-TransH | .446 | .614 | .728 | .728 |
| m-DistMult | .460 | .635 | .784 | .845 |
| m-CP | .392 | .560 | .752 | .837 |
| HypE | .492 | .650 | .804 | .856 |
| HSimplE | .472 | .649 | .798 | .855 |
| BoxE(u) | .553 | .711 | .837 | .895 |
| BoxE(a) | **.560** | **.722** | **.844** | **.898** |

This is primarily due to the natural extensibility of BoxE to non-uniform and higher arity. Indeed, BoxE defines unique boxes for every arity position, enabling a more natural representation of entity sets at every relation position. By contrast, all other models represent all relations with identical embedding structures, which can bottleneck the learning process, in particular when arities vary. Furthermore, the inductive capacity of BoxE also naturally extends to higher arities as a result of its structure, namely for higher-arity hierarchy, intersection, and mutual exclusion, which further improves its learning ability in this setting.

## 6.3 Rule injection

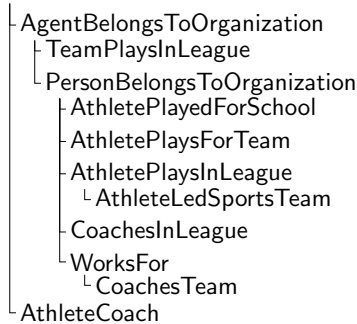

Figure 3: The SportsNELL ontology.

In this experiment, we investigate the impact of rule injection on BoxE performance on the SportsNELL dataset, a subset of NELL [26] with a known ontology, shown in Figure 3. We also consider the dataset SportsNELL$^C$, which is precisely the logical closure of the SportsNELL dataset w.r.t. the given ontology (i.e., completion of SportsNELL under the rules).

We compare plain BoxE with BoxE injected with the SportsNELL ontology, denoted BoxE+RI. We train both models for 2000 epochs on a random subset (90%) of SportsNELL. First, we evaluate both models on all remaining facts from SportsNELL$^C$, which we refer to as the *full evaluation set*, to measure the effect of rule injection. Second, we evaluate both models on a subset of the full evaluation set, only consisting of facts that are *not* directly deducible via the ontology from the training set (i.e., eliminating all inferences that can be made by a rule-based approach alone). This subset, which we call the *filtered evaluation set*, thus carefully tests the impact of rule injection on model inductive capacity.

**Results.** The results on both evaluation datasets are shown in Table 4. On the full evaluation dataset, BoxE+RI performs significantly better than BoxE. This shows that rule injection clearly improves the performance of BoxE. Importantly, this performance improvement cannot solely be attributed to the facts that can be deduced directly from the training set (with the help of the rules), as BoxE+RI performs much better than BoxE also over the filtered evaluation set. These experiments suggest that rule injection improves the inductive bias of BoxE, by enforcing all predictions to also conform with the given set of rules, as required. Intuitively, all predictions get amplified with the help of the rules, a very desired property, as many real-world KBs have an associated schema, or a simple ontology.

Table 4: Rule injection experiment results on the SportsNELL full and filtered evaluation sets.

| Model | Full Set | | | Filtered Set | | |
|---|---|---|---|---|---|---|
| | MR | MRR | H@10 | MR | MRR | H@10 |
| BoxE | 17.4 | .577 | .780 | 19.1 | .713 | .824 |
| BoxE+RI | **1.74** | **.979** | **.997** | **5.11** | **.954** | **.984** |

While allowing to amplify predictions, rule injection can potentially lead to poor performance with existing metrics. Indeed, if a model mostly predicts wrong facts, these would lead to further wrong conclusions due to rule application. Hence, a low-quality prediction model can find its performance further hindered by rule injection, as false predictions create yet more false positives, thereby lowering the rank of any good predictions in evaluation. Therefore, rule injection must be complemented with models having good inductive capacity (for sparser and simpler datasets) and expressiveness (for more complex and rich datasets), such that they yield high-quality predictions in all data settings [35].

## 7 Summary

We presented BoxE, a spatio-translational model for KBC, and proved several strong results about its representational power and inductive capacity. We then empirically showed that BoxE achieves state-of-the-art performance both for KGC, and on higher-arity and different-arity KGC. Finally, we empirically validated the impact of rule injection, and showed it improves the overall inductive bias and capacity of BoxE. Overall, BoxE presents a strong theoretical backbone for KBC, combining theoretical expressiveness with strong inductive capacity and promising empirical performance.

## Acknowledgments

This work was supported by the Alan Turing Institute under the UK EPSRC grant EP/N510129/1, the AXA Research Fund, and by the EPSRC grants EP/R013667/1 and EP/M025268/1. Ralph Abboud is funded by the Oxford-DeepMind Graduate Scholarship and the Alun Hughes Graduate Scholarship. Experiments for this work were conducted on servers provided by the Advanced Research Computing (ARC) cluster administered by the University of Oxford.

## Broader Impact

The representation and inference of knowledge is essential for humanity, and thus any improvements in the quality and reliability of automated inference methods can significantly support endeavors in several application domains. This work provides a means for dealing with incomplete knowledge, and offers users to complete their knowledge bases with the help of automated machinery. The model predictions rely mostly on interpretable and explainable logical patterns, which makes it easier to analyze the model behavior. Furthermore, this work enables safely injecting background rules when completing knowledge bases, and this safety is of great value in settings where inferred knowledge is critical (e.g., completing medical knowledge bases). This work thus also provides a logically grounded approach that improves the quality of predictions and completions in safety-critical settings. The ability of the proposed model to naturally handle more general knowledge bases (beyond knowledge graphs) could also unlock the use of knowledge base completion technologies on important knowledge bases which were previously ignored.

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
