[Supplementary Material]

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

# A  Runtime and Space Complexity of BoxE

**Runtime.** For any fact $r(e_1, \ldots, e_n)$, we can compute the entity representations $e^{r(e_1, \ldots, e_n)}$, in time $O(nd)$, by first computing $\sum_{1 \leq i \leq n} b_i$ in $O(nd)$, then subtracting $b_i$ from the overall sum for every entity $e$ and finally adding the base position $e$, resulting in $3n$ $d-$dimensional addition/subtraction operations. The distance function dist runs in $O(d)$ for every box and entity, as it involves a fixed number of $d-$dimensional operations. Thus, running dist for all $n$ positions yields a running time of $O(nd)$. Hence, BoxE scoring runs in $O(nd)$ overall. This implies that BoxE scales linearly with the arity of the relations in a KB, and thus can be applied to this setting with minimal computational overhead. Assuming that $n$ is bounded, as is the case for KGs, BoxE runs in *linear time* with respect to dimensionality $d$.

**Space complexity.** In terms of space complexity, BoxE stores $2$ $d-$dimensional vectors per entity $e$, namely its base position $e$ and bump $b$, and stores $2$ $d-$dimensional vectors per box, denoting its lower and upper corners. Hence, for a KB with $|\mathbf{E}|$ entities and $|\mathbf{R}|$ relations with arity $n$, BoxE requires $(|\mathbf{E}| + n|\mathbf{R}|)d$ parameters.

# B  Proof of Theorem 5.1 (Full Expressiveness)

We first prove the result for knowledge graphs, and then show how this can be lifted to arbitrary knowledge bases with higher-arity relations.

The result is shown by induction. We start with a base case where the KG $G$ contains all facts from the universe as true facts, and subsequently prove in the induction step that a BoxE model with $d = |\mathbf{E}||\mathbf{R}|$ can make any arbitrary fact in $G$ false without affecting the correctness of other facts. In this induction step, facts are made false by pushing the representation of a single entity in the fact outside its corresponding relation box at a specific dimension, and modifying the remaining embeddings in the model to prevent a change in the truth value of any other fact.

Let us assume without loss of generality that all relations and entities are indexed. Specifically, we consider relations $r_i \in \mathbf{R}$, and entities $e_j \in \mathbf{E}$, where $0 \leq i \leq |\mathbf{R}| - 1$, and $0 \leq j \leq |\mathbf{E}| - 1$. We consider $d$-dimensional embedding vectors $v$ with $d = |\mathbf{E}||\mathbf{R}|$, and write $v(i, j)$ to refer to the vector index $i|\mathbf{E}| + j$. Intuitively, in our construction, the sequence of indices $v(i, 0), \ldots, v(i, |\mathbf{E}| - 1)$ corresponds to a "chunk" reserved for the relation $r_i$.

**Base case:** We initialize the KG $G$ as the whole universe, i.e., the set of all possible facts over a given vocabulary. BoxE can trivially express $G$, by simply setting all entity and bump vectors to $0$, and all boxes as the unit box centered at $0$.

**Induction step:** In this step, we consider a true fact $r_i(e_j, e_k)$, and make this fact false without affecting the remainder of G. This can be done as follows:

Step 1. Increment $b_j(i, k)$ by a value $C$, such that:
$$e_k(i, k) + b_j(i, k) + C > u_i^{(2)}(i, k).$$

Step 2. Decrement all entity embeddings except that of $e_k$ by $C$ at dimension $(i, k)$:
$$\forall \, k' \neq k, e_{k'}(i, k) = e_{k'}(i, k) - C.$$

Step 3. For the relation $r_i$, grow the head box by $C$ at dimension $(i, k)$ both upwards and downwards, and grow the tail box downwards by $C$ in this dimension:
$$l_i^{(1)}(i, k) = l_i^{(1)}(i, k) - C,$$
$$u_i^{(1)}(i, k) = u_i^{(1)}(i, k) + C,$$
$$l_i^{(2)}(i, k) = l_i^{(2)}(i, k) - C.$$

Step 4. For all other relations $r_x \in \mathbf{R}, x \neq i$, grow all boxes by $C$ at dimension $(i, k)$ in both directions, that is, for $\beta \in \{1, 2\}$:
$$l_x^{(\beta)}(i, k) = l_x^{(\beta)}(i, k) - C,$$
$$u_x^{(\beta)}(i, k) = u_x^{(\beta)}(i, k) + C.$$

Observe first that Step 1 makes $r_i(e_j, e_k)$ false, by pushing $e_k^{r_i(e_j,e_k)}$ outside of $r_i^{(2)}$ at dimension $(i, k)$ from above. This flips the truth value of $r_i(e_j, e_k)$, as required.

We now show that the results of Steps 1 & 2, combined with the changes to relation boxes made in Steps 3 & 4, which affect facts involving $r_i$ and other relations respectively, preserve the correctness/falsehood of all facts other than $r_i(e_j, e_k)$. To this end, we consider any possible fact $F = r_{i'}(e_{j'}, e_{k'})$ from the KG, and analyze the effect of the induction step at the head and tail of the fact. We need to consider the following cases:

Case 1. **The fact $F$ is true:** To verify that $F = r_{i'}(e_{j'}, e_{k'})$ remains true after the inductive step, we analyze both the head entity $e_{j'}^F$ and the tail entity $e_{k'}^F$.

   (a) **Head entity:** Observe that (i) $e_{j'}^F$ can change by at most $C$ following Steps 1 & 2, and (ii) all relation head boxes are grown by $C$ in both directions in Steps 3 & 4. These together imply that $e_{j'}^F \in r^{(1)}$ is guaranteed to hold provided that it was true before the induction step.

   (b) **Tail entity:** If $e_{k'} \neq e_k$, then $e_{k'}^F$ is not changed if $e_{j'} = e_j$, and decremented by $C$ at dimension $(i, k)$ otherwise. Hence, the changes to both $r_i^{(2)}$ and $r_x^{(2)}, x \neq i$ in Steps 3 & 4 are sufficient to maintain $e_{k'} \in r^{(2)}$. Conversely, if $e_{k'} = e_k$, then $e_t^F$ is unchanged when $e_{j'} \neq e_j$, and thus $e_{k'} \in r^{(2)}$ still holds. Otherwise, when $e_{j'} = e_j$, $e_{k'}^F$ is incremented by $C$, which, for $r = r_i$, makes $F$ false, as required, and for $r \neq r_i$, still keeps $e_{k'} \in r^{(2)}$, as all other tail boxes are grown upwards by C.

   Hence, for any true fact in $G$, except the fact $r_i(e_j, e_k)$, we conclude that $e_{j'} \in r^{(1)}$ and $e_{k'} \in r^{(2)}$ continues to hold after the induction step, as required.

Case 2. **The fact $F$ is false:** To verify that $F = r_{i'}(e_{j'}, e_{k'})$ remains false, after the inductive step, we again consider the head and tail entities.

   (a) **Head entity:** By construction, all false facts $r_{i'}(e_{j'}, e_{k'})$ satisfy the inequality

   $$e_{k'}^{r_{i'}(e_{j'},e_{k'})}(i', k') > u_{i'}^{(2)}(i', k'),$$

   and any changes to $e_{j'}^F$ do not affect this inequality.

   (b) **Tail entity:** If $e_{k'} \neq e_k$, then $F$ verifies $e^F(i', k') > u_{i'}^{(2)}(i', k')$, where $k' \neq k$. This inequality continues to hold regardless of the changes to $e_{k'}^F(i, k)$. Otherwise, if $e_{k'} = e_k$, and $r_{i'} = r_i$, then $e_{j'} \neq e_j$, as $F$ is initially false, and $r_i(e_j, e_k)$ is initially true. Furthermore, since $e_{j'} \neq e_j$, $e_{k'}^F$ is unchanged, which maintains the falsehood inequality. Finally, if $r_{i'} \neq r_i$, then the falsehood inequality for $F$ holds at a dimension different than $(i, k)$. Therefore, none of the changes in the induction step affect this inequality.

   Hence, all false facts in $G$ remain false after the induction step, as required.

Thus, the induction step can make any true fact $r_i(e_j, e_k)$ in G false in a BoxE model with $d = |\mathbf{E}||\mathbf{R}|$ without affecting the remainder of the facts in G. Hence, all fact configurations are possible and expressible by such a BoxE model, and this model is fully expressive, as required.

This proof can be generalized to higher-arity knowledge bases. Indeed, for a maximum arity of $n$, a dimensionality $d = |\mathbf{E}|^{n-1}|\mathbf{R}|$ is needed for full expressiveness. All proof steps shown above would remain the same, except that (i) we define a higher-arity indexing function $(\theta_1, \theta_2, ..., \theta_K)$, which refers to vector index $\sum_{a=1}^{n} |\mathbf{E}|^{n-a}\theta_i$, and (ii) grow boxes for $r_i$ and all other $r_x$ at positions 3 and onwards in both Steps 3 and 4, in addition to position 1, by $C$ in both directions (while the changes at position 2 remain the same).

Finally, we note that the proof can be trivially extended to knowledge bases with non-uniform arities (i.e., KBs containing relations with different arities) by introducing extra parameters to relations of lower arity, and setting the correctness of the $n-$arity facts solely based on the original facts. Hence, BoxE is a fully expressive model for general KBs containing both *distinct* and *large* relation arities.

# C Proof of Theorem 5.2 (Inference Patterns and Generalizations)

We start by more explicitly reformulating Definition 5.1 in the main body of the paper.

**Definition C.1.** *Generalized inference patterns are defined as follows:*

- *A* symmetry rule *is of the form* $r_1(x, y) \Rightarrow r_1(y, x)$, *where* $r_1 \in \mathbf{R}$. *A generalized symmetry pattern is a finite set of symmetric rules over* $\mathbf{R}$.

- *An* anti-symmetry rule *is of the form* $r_1(x, y) \Rightarrow \neg r_1(y, x)$, *where* $r_1 \in \mathbf{R}$. *A generalized anti-symmetry pattern is a finite set of anti-symmetric rules over* $\mathbf{R}$.

- *An* inversion rule *is of the form* $r_1(x, y) \Leftrightarrow r_2(y, x)$, *where* $r_1 \neq r_2 \in \mathbf{R}$. *A generalized inversion pattern is a finite set of inverse rules over* $\mathbf{R}$.

- *A* composition rule *is of the form* $r_1(x, y) \wedge r_2(y, z) \Rightarrow r_3(x, z)$, *where* $r_1 \neq r_2 \neq r_3 \in \mathbf{R}$. *A generalized composition pattern is a finite set of composition rules over* $\mathbf{R}$.

- *A* hierarchy rule *is of the form* $r_1(x, y) \Rightarrow r_2(x, y)$, *where* $r_1 \neq r_2 \in \mathbf{R}$. *A generalized hierarchy pattern is a finite set of hierarchy rules over* $\mathbf{R}$.

- *An* intersection rule *is of the form* $r_1(x, y) \wedge r_2(x, y) \Rightarrow r_3(x, y)$, *where* $r_1 \neq r_2 \neq r_3 \in \mathbf{R}$. *A generalized intersection pattern is a finite set of intersection rules over* $\mathbf{R}$.

- *A* mutual exclusion rule *is of the form* $r_1(x, y) \wedge r_2(x, y) \Rightarrow \bot$, *where* $r_1 \neq r_2 \in \mathbf{R}$. *A generalized mutual exclusion pattern is a finite set of mutually exclusive rules over* $\mathbf{R}$.

*Every generalized inference pattern defines a trivial rule language, consisting of a single type of rule. We define more general rule languages, by taking the union of different types of rules. A rule language $\mathcal{L}$ is defined in terms of the types of rules that are allowed in the language.*

**Remark.** The requirement for setting the relations to be distinct is due to the existing conventions in the literature. This may appear somewhat unintuitive, but it is required to study the rules in isolation, i.e., a composition rule without this requirement can express transitivity by defining $r_1(x, y) \wedge r_2(y, z) \Rightarrow r_1(x, z)$ which cannot be captured by models that do capture composition. Nevertheless, this assumption does not lead to loss of generality for our study of generalized inference patterns, since we are allowed to use many rules, which in turn, can easily simulate cases that are excluded. For instance, the following rules: $r_1(x, y) \wedge r_2(y, z) \Rightarrow r_3(x, z)$, and $r_3(x, y) \Rightarrow r_1(x, y)$ together simulate the transitivity rule given above.

We prove the statements in Theorem 5.2 in two seperate parts. First, we prove the results given for BoxE from Table 1, and then we show the results given for the other models from Table 1.

## C.1 Proof of Theorem 5.2: BoxE

We show that each generalized inference pattern can be captured by BoxE except for the composition pattern. For the latter, we argue why BoxE cannot capture this explicitly, as an inference pattern.

**Generalized intersection.** We first introduce the concept of *boxicity*. Let $G = (V, E)$ be a graph, where $V$ is the set of nodes, and $E$ is the set of edges. The *boxicity* of $G$ is the minimum embedding dimension in which $G$ can be represented as an intersection of axis-aligned boxes, such that (i) every box corresponds to a specific node, (ii) boxes intersect iff an edge connects their respective nodes [31]. It has been shown that the boxicity of a graph with $p$ edges is $O(\sqrt{p \cdot \log(p)})$ [5]. This implies that, given a graph $G$, where every relation $r \in \mathbf{R}$ is represented as a node in the graph, and every edge between them represents an intersection, any finite combination of intersections between relations can be represented in a finite-dimensional vector space of worst-case dimensionality $O(|\mathbf{R}|\sqrt{\log(|\mathbf{R}|)})$.

For a given knowledge graph, we define a *relation intersection graph*. That is, for every relation $r \in \mathbf{R}$, we define two nodes, corresponding to its head and tail boxes, and then set edges in the graph based on desired intersections between relation boxes, which are dictated by intersection rules.

Prior to encoding rules into relation intersection graph edges, we first compute the *deductive closure* of the set of intersection rules. In other words, we check whether any rule of the form $r_i(x, y) \Rightarrow$

$r_j(x, y)$, or $r_i(x, y) \land r_j(x, y) \Rightarrow r_k(x, y)$ for any $i, j, k$ can be entailed from the given set of rules, and keep adding new rules to this initial set, until no more rules can be deduced. That is, we compute the logical closure of the initial set. This allows us to make all possible intersections between relations explicit.

Then, we map all rules in the computed deductive closure to edges as follows:

- For every intersection rule $r_1(x, y) \land r_2(x, y) \Rightarrow r_3(x, y)$, we set edges between the node corresponding to the head of $r_3$ and those of $r_1$ and $r_2$, with the same done for tail nodes.
- For every deduced hierarchy rule $r_1(x, y) \Rightarrow r_2(x, y)$, we set edges between the head nodes of $r_2$ and $r_1$, with the same done for tail nodes.

With the resulting relation intersection graph $G$, we have encoded necessary conditions for all rules to hold, namely that relations whose intersections are contained in other relations intersect with these relations. We now leverage the boxicity argument, and show that there exists a box configuration of finite dimensionality capturing all the intersections encoded in $G$. This box configuration captures all intersections needed between the respective boxes for the rules to hold, but is not necessarily sufficient to capture hierarchies and box containment. Hence, we modify the aforementioned box configuration using a procedure, which we apply iteratively over every intersection rule, such that the final configuration provably captures all rules, without capturing additional undesired rules.

Our box reconfiguration procedure is as follows:

1. Iterate over every intersection rule $r_1(x, y) \land r_2(x, y) \Rightarrow r_3(x, y)$:
   (a) If the $r_3$ head and tail boxes do not contain the head and tail box intersections $r_1 \cap r_2$, then we grow these $r_3$ boxes by the minimum possible amount to make this condition hold and establish the rule. In other words, we grow the $r_3$ boxes at every position to equal the boundary of either the $r_1$ boxes or the $r_2$ boxes at the dimensions where the rule does not hold due to $r_1$ or $r_2$. This growth operation preserves all existing edges in $G$, and does not force new intersections, as all forced intersections due to rule capturing are already encoded by the existing edges.
   (b) Following Part (a), the growth of $r_3$ can violate another rule in the set, in particular if $r_3$ is in the body of this rule. Hence, when any $r_3$ boxes are grown in Part (a), check all other intersection rules in the rule set: If the change in $r_3$ makes a rule no longer hold (i.e., the rule was captured prior to growing $\boldsymbol{r_3}$ and no longer is), then recursively call this procedure for this rule.

We now show that this procedure is correct, and then prove that it terminates, particularly with respect to the number of recursive calls made. First, we note that, following a successful iteration on a given rule, a rule is successfully captured (Part (a)), and no other rules are violated in the process (Part (b)). Thus, the final configuration returned by this procedure over the initial boxicity-given configuration returns a valid BoxE configuration. In particular, this configuration captures all and only the provided patterns within the deductive closure, which includes the original intersection rules. Furthermore, growing $r_3$ boxes in the configuration to satisfy an intersection rule does not induce any rules outside their deductive closure. Indeed, when $r_3$ boxes are grown, they are only grown in dimensions where they fail to capture $\boldsymbol{r_1} \cap \boldsymbol{r_2}$. Thus, the procedure can only make $\boldsymbol{r_3}$ intersect with boxes that intersect with $\boldsymbol{r_1}$ or $\boldsymbol{r_2}$. As a result, the procedure can only force intersections between boxes within the deductive closure of the rule set.

It now remains to show that this procedure terminates, and thus that a configuration of this kind indeed can be found. In particular, we study the maximal number of recursive calls needed. Consider a rule $\rho : r_1(x, y) \land r_2(x, y) \Rightarrow r_3(x, y)$, where $r_3$ boxes are grown. For simplicity, we only consider a single box for $r_3$, i.e., a unique arity position, as the analysis is analogous at every arity position. We define *boundaries* as being the lower and upper limits of a box at every dimension. Thus, a $d$-dimensional box has $2d$ boundaries. Therefore, in our binary BoxE configuration with $|\mathbf{R}|$ relations and $d = O(|\mathbf{R}|\sqrt{\log |\mathbf{R}|})$, there are $O(|\mathbf{R}|^2 \sqrt{\log |\mathbf{R}|})$ boundaries. For our analysis, we are interested in the number of *distinct* boundaries in our configuration.

We now consider the effect of an application of a call to Part (a) of the procedure on the number of distinct boundaries. If $\rho$ is already captured, then no action is needed. Otherwise, $\boldsymbol{r_3}$ needs to be grown. Hence, in this scenario, there exists at least one dimension in which the lower (resp., upper)

boundary of $r_3$ is strictly higher (resp., lower) than the maximum (resp., minimum) lower (resp., upper) bound of either $r_1$ or $r_2$. Therefore, when $r_3$ is grown, the value of the problematic bound(s) at this dimension is made equal to the corresponding bound(s) of $r_1$ or $r_2$. As a result, the number of distinct boundaries is guaranteed to strictly drop by at least 1 following any growth operation.

Furthermore, we consider the recursive calls made in Part (b), after any growth to $r_3$. Observe that recursion is only called when the change to $r_3$ exclusively makes the checked rule false. This condition ensures that all recursive calls are made only when the growing of the rule head boxes, in this case $r_3$, is the exclusive cause for rule violation, and so eliminates all other possible causes of violation such that they are handled only when the outer loop iterating reaches the corresponding rule, and thus greatly simplifies the analysis. Finally, we observe that box growth can only be triggered when distinct boundaries exist. Hence, when the number of distinct boundaries drops to its (highly pessimistic and loose) minimum possible value of 1, no more recursive calls can be made. This observation, combined with the earlier finding that every box growth strictly reduces the number of distinct boundaries by at least 1, implies that the number of recursive calls in this procedure is upper bounded by $O(|\mathbf{R}|^2 \sqrt{\log |\mathbf{R}|})$. Hence, this procedure terminates, and a BoxE configuration capturing generalized intersections exists.

**Generalized hierarchy.**  The proof for generalized intersection immediately applies to generalized hierarchies.

**Generalized symmetry.**  The symmetry inference pattern is a single-relation pattern, and can appear at most once per relation. Symmetry can be easily captured for a relation $r$ by setting $r^{(1)}$ and $r^{(2)}$ to be identical boxes. This can be independently done for any relation, and thus BoxE captures generalized symmetry.

**Generalized anti-symmetry.**  Analogously to generalized symmetry, anti-symmetry is a single-relation pattern. This pattern is captured by setting $r^{(1)}$ and $r^{(2)}$ to be disjoint for every anti-symmetric $r$. Therefore, BoxE captures generalized anti-symmetry.

**Generalized inversion.**  An inversion pattern $r_1(x, y) \Leftrightarrow r_2(y, x)$ can be captured by setting $r_1^{(1)}$ and $r_2^{(2)}$, as well as $r_1^{(2)}$ and $r_2^{(1)}$, to be identical boxes. This box sharing between inverse relations can easily be extended to any arbitrary set of inversion rules.

**Generalized mutual exclusion.**  It is sufficient to observe that there exists a BoxE configuration for any arbitrary set of mutual exclusion rules due to the boxicity argument: simply consider a graph $G$ with no edges connecting mutually exclusive relations. A simpler argument can be given directly: generalized mutual exclusion can be achieved by making one of the relation boxes (head, or tail) disjoint in a fixed-dimensional space.

**(Generalized) composition.**  Consider the composition pattern $r_1(x, y) \wedge r_2(y, z) \rightarrow r_3(x, z)$. In this pattern, we see that the entity that will appear in lieu of variable $x$ will be bumped differently in every atom, as it appears with different entities. More concretely, if we replace variables $x, y, z$ with entities $e_1, e_2, e_3$ respectively, then $e_1^{r_1} = e_1 + b_2$ and $e_1^{r_3} = e_1 + b_3$. We can also view bumps as equivalently applying to boxes, i.e., instead of $e_1 + b_2 \in r_1^{(1)}$, we write $e_1 \in r_1^{(1)} - b_2$. Hence, it is equivalent to view BoxE as bumping relation boxes in the opposite direction.

Now, we can see that $r_1^{(1)}$ is bumped by $-b_2$, whereas $r_3^{(1)}$ is bumped by $-b_3$. Therefore, since bumps are entity-specific and unknown a priori since the bump stems from an abstract variable, one cannot analyze the relative positions of $r_1^{(1)}$ and $r_3^{(1)}$ and draw conclusions. By contrast, all other captured rules in BoxE are such that relation boxes corresponding to the same variable are bumped identically, which in effect neutralizes the effect of bumping and enables the capturing of the patterns. Hence, translational bumps, which allow BoxE to be fully expressive, prevent BoxE from capturing compositions.

## C.2    Proof of Theorem 5.2: Other models

In what follows, we generally define KGC embedding models such that every KG entity is represented by a vector in $\mathbb{R}^d$, and every relation defines two map functions $r_h, r_t : \mathbb{R}^d \to \mathbb{R}^d$, which apply to head and tail embeddings, respectively. We further define the relation scoring function over a KG triple $s_r : \mathbb{R}^d \times \mathbb{R}^d \to \mathbb{R}$ as a map from entity pair representations following the application of $r_h$ and $r_t$ to a real-valued score.

### C.2.1    Translational models: TransE and RotatE

We note that some of the results stated below are taken from the literature, but we included them nevertheless for completeness. The novel results are given for the generalized inference patterns.

For translational models, $r_h(\boldsymbol{e_1})$ encodes the translation (resp. rotation) operation, $r_t(\boldsymbol{e_2}) = \boldsymbol{e_2}$, and $s_r(e_1, e_2) = \|r_t(\boldsymbol{e_2}) - r_h(\boldsymbol{e_1})\|$.

**Hierarchy.**    Let $\mathcal{M}_r = s_r^{-1}([0, \epsilon])$, where $s_r^{-1}$ is the inverse map of $s_r$, be the subset of embedding pairs $(v, w) \in \mathbb{R}^d \times \mathbb{R}^d$ such that $s_r(v, w) \leq \epsilon$, i.e., the decision region of the relation $r$ with margin $\epsilon$. As a result, $r_1(x, y) \Rightarrow r_2(x, y)$ holds iff $\mathcal{M}_1 \subset \mathcal{M}_2$. In TransE (resp., RotatE), $(\boldsymbol{e_1}, \boldsymbol{e_2}) \in \mathcal{M}_r$ if $\boldsymbol{e_1} + \boldsymbol{r} - \boldsymbol{e_2} \in D_\epsilon(0)$ (resp., $\|\boldsymbol{e_1} \circ \boldsymbol{r} - \boldsymbol{e_2}\| \in D_\epsilon(0)$), where $D_\epsilon(0)$ is the disk of center 0 and radius $\epsilon$. Since it is necessary that $M_1 \subset M_2$, we require that the disk $D_{1,\epsilon}(\boldsymbol{e_1} + \boldsymbol{r_1})$ (resp. $D_{1,\epsilon}(\boldsymbol{e_1} \circ \boldsymbol{r_1})$) and radius $\epsilon$ is contained in the corresponding disk $D_2$, defined analogously using $r_2$. Since $D_1$ and $D_2$ have the same margin-induced radius, this is only possible if $r_1 = r_2$, effectively enforcing relation equivalence. Thus, neither translational model can capture hierarchies.

**Intersection.**    A model can represent the intersection pattern $r_1(x, y) \wedge r_2(x, y) \Rightarrow r_3(x, y)$ if $\mathcal{M}_1 \cap \mathcal{M}_2 \subset \mathcal{M}_3$. In TransE and RotatE, this is satisfied if $r_3$ lies in the centre of the disk intersection of $D_\epsilon(r_1)$ and $D_\epsilon(r_2)$, thus both models capture intersection. However, both models clearly fail to capture generalized intersection. In particular, if we consider rules $r_1(x, y) \wedge r_2(x, y) \Rightarrow r_3(x, y)$ and $r_3(x, y) \wedge r_2(x, y) \Rightarrow r_1(x, y)$, the rule $r_2(x, y) \Rightarrow r_1(x, y)$ is logically implied. But this is a hierarchy rule that clearly cannot be captured by either model. Hence, TransE and RotatE cannot capture generalized intersections.

**Symmetry.**    In TransE, $r(x, y) \Rightarrow r(y, x)$ holds iff $\boldsymbol{r} = 0$, which implies that $r$ is reflexive. Thus, TransE does not capture symmetry. In contrast, in RotatE, symmetry is captured iff $r = \{\pm k\pi\}^d$, $k \in \mathbb{N}$, i.e., a rotation vector consisting exclusively of multiples of $\pi$. Symmetry is a single-relation pattern, and thus multiple rules, affecting different relations, can be captured independently. Hence, RotatE captures generalized symmetry.

**Anti-symmetry.**    In TransE, a relation $r$ is anti-symmetric iff $\|\boldsymbol{r}\| \geq \epsilon$. The result for RotatE is proven in the original work [37]. As anti-symmetry is a single-relation pattern, it can be applied independently across all relations. Thus, both TransE and RotatE capture generalized anti-symmetry.

**Inversion.**    For both TransE and RotatE, inversion holds iff $\boldsymbol{r_1} = -\boldsymbol{r_2}$. However, whereas RotatE can capture generalized inversion through repeated application of the earlier equation across all inversion rules, since it can handle any deduced symmetry results, TransE cannot. More concretely, consider the rule set $r_1(x, y) \Leftrightarrow r_2(y, x)$, $r_2(x, y) \Leftrightarrow r_3(y, x)$, $r_3(x, y) \Leftrightarrow r_1(y, x)$. This rule set implies $r_1(x, y) \Leftrightarrow r_1(y, x)$, which RotatE can capture, but which TransE cannot. More generally, generalized inversion rules can yield symmetry rules, and thus only RotatE can capture generalized inversion.

**Mutual exclusion.**    To capture mutual exclusion between relations $r_1$ and $r_2$, the model must satisfy $\mathcal{M}_1 \cap \mathcal{M}_2 = \varnothing$. In TransE, this holds iff $\|\boldsymbol{r_1} - \boldsymbol{r_2}\| \geq 2\epsilon$. Analogously, for RotatE, this holds if $|\boldsymbol{r_i} - \boldsymbol{r_j}| \geq \arcsin(2\epsilon)$ at every dimension and all node embeddings have a norm of at least 1. Such constructions can be set up for arbitrarily many mutual exclusion pairs, through decreasing $\epsilon$ or increasing the magnitude of embeddings. Thus, both TransE and RotatE can capture generalized mutual exclusions.

**Composition.** For TransE (resp., RotatE), two relations $r_1$ and $r_2$ compose a third relation $r_3$ iff $\boldsymbol{r_1} + \boldsymbol{r_2} = \boldsymbol{r_3}$ (resp., $\boldsymbol{r_1} \circ \boldsymbol{r_2} = \boldsymbol{r_3}$). On the other hand, both fail to capture generalized compositions. In particular, for the rules $r_1(x,y) \wedge r_2(y,z) \Rightarrow r_3(x,z)$ and $r_1(x,y) \wedge r_4(y,z) \Rightarrow r_3(x,z)$, both models force $r_2 = r_4$ (In RotatE, the equality is modulo $2\pi$).

### C.2.2 Bilinear models: DistMult, ComplEx, TuckER

TuckER is shown to subsume DistMult and ComplEx [1], so all positive results for either ComplEx and DistMult automatically follow for TuckER. Hence, these positive results for TuckER are omitted from the presentation. Analogously, when negative results are shown for TuckER, they automatically propagate to DistMult and ComplEx.

We now formally introduce TuckER. TuckER learns a tensor $\mathcal{W} \in \mathbb{R}^{d_e \times d_r \times d_e}$, $d_e, d_r \in \mathbb{N}$, a vector $\boldsymbol{e} \in \mathbb{R}^{d_e}$ for every entity, and a vector $\boldsymbol{r} \in \mathbb{R}^{d_r}$ for every relation, and $s_r(\boldsymbol{e_1}, \boldsymbol{e_2}) = \mathcal{W} \cdot \boldsymbol{e_1} \cdot \boldsymbol{r} \cdot \boldsymbol{e_2}$. For ease of notation, we define $v_{1,r} = \mathcal{W} \times \boldsymbol{e_1} \times \boldsymbol{r}$. The scoring function can then be written as $s_r(\boldsymbol{e_1}, \boldsymbol{e_2}) = v_{r,1} \cdot \boldsymbol{e_2}$. Given a head entity $e_1$ and a relation $r$, we define the space $A_{1,r} = \{\boldsymbol{x} \in \mathbb{R}^{d_e} \mid v_{r,1} \cdot \boldsymbol{x} \geq \epsilon\}$.

**Hierarchy.** For bilinear models, it has been shown that individual hierarchies can be captured, but not generalized hierarchies [14]. In particular, to satisfy the rules $r_1(x,y) \Rightarrow r_3(x,y)$ and $r_2(x,y) \Rightarrow r_3(x,y)$ simultanously, bilinear models must set either $r_1(x,y) \Rightarrow r_2(x,y)$ or $r_2(x,y) \Rightarrow r_1(x,y)$.

**Intersection.** We show that TuckER cannot capture intersections. In TuckER, a rule of the form $r_1(x,y) \wedge r_2(x,y) \Rightarrow r_3(x,y)$ holds iff $A_{r_2,1} \cap A_{r_1,1} \subset A_{r_3,1}$, $\forall \boldsymbol{e} \in \mathbb{R}^{d_e}$. This is true iff $v_{r_1,1}, v_{r_2,1}, v_{r_3,1}$ are colinear, and thus that $r_1, r_2$, and $r_3$ are colinear. However, this also implies that either $r_1(x,y) \Rightarrow r_2(x,y)$, or $r_2(x,y) \Rightarrow r_1(x,y)$. Hence, TuckER fails to capture intersections.

**Symmetry.** ComplEx captures symmetry patterns by having real-only embedding matrices for its relations. DistMult is inherently symmetric by construction. Since symmetry is a single-relation pattern, multiple symmetries can be independently captured, and thus all three models can capture generalized symmetry.

**Anti-symmetry.** DistMult cannot capture anti-symmetry, as it is inherently a symmetric model. ComplEx captures anti-symmetry by having imaginary-only embedding matrices for its relations. Analogously to symmetry, anti-symmetry is also a single-relation pattern, and thus ComplEx (and TuckER) can capture generalized anti-symmetry.

**Inversion.** It is known that DistMult cannot capture inversions, while ComplEx can [37]. Generalized inversion can also be captured in ComplEx, as symmetry, the only other type of rule deducible from multiple inversions, is also captured by ComplEx.

**Mutual exclusion.** In TuckER, two relations $r_1$ and $r_2$ are mutually exclusive iff $r_1 = -r_2$. This implies TuckER can capture mutual exclusion, but cannot capture generalized mutual exclusions. In particular, to satisfy $r_1(x,y) \wedge r_2(x,y) \Rightarrow \bot$ and $r_1(x,y) \wedge r_3(x,y) \Rightarrow \bot$, TuckER forces $r_2 = r_3$.

**Composition.** It is shown that both ComplEx and DistMult cannot capture composition patterns [37, 14]. Furthermore, it is also known that relation maps must be bijective to be able to represent composition [37]. This is not the case in TuckER, as relations are surjective maps from $\mathbb{R}^{d_e \times d_r}$ to $\mathbb{R}^{d_e}$, and linear bijections between vector spaces are only possible with the same dimensionality. Hence, TuckER also cannot capture compositions.

## D  Proof of Theorem 5.3 (Inference Patterns as Rule Languages)

We show that a BoxE model of dimensionality $d = O(|\mathbf{R}|^2)$ captures the rule language specified in Theorem 5.3. This is achieved by leveraging the ideas from the generalized inference patterns proof in Appendix C. Indeed, our existence proof also builds on the boxicity argument used in this proof.

Let $S$ be a set of rules, and let $S_p$, $S_a$, and $S_m$ be subsets of $S$, where $S_p$ consists of hierarchy, symmetry, inversion, and intersection rules, $S_a$ consists of anti-symmetry rules, and $S_m$ consists of mutual exclusion rules. We first show that rules from $S_p \cup S_a$ can be captured, then extend this to additionally capture $S_m$.

**Step 1: Defining the relation intersection graph.** We define a set of $2|\mathbf{R}|$ nodes, where every relation is encoded with 2 nodes for its head and tail boxes. We now constrain this graph to eventually capture all rules in $S_p$. First, we capture all symmetry and inversion rules as follows:

1. **Symmetry:** For every symmetry rule, we combine the corresponding head and tail nodes of a relation $r$ to a single node. In other words, a single relation $r$ is made symmetric by encoding both $r^{(1)}$ and $r^{(2)}$ with one same node. This encoding enforces that the head and tail boxes of $r$ are identical, and thus that $r$ is indeed symmetric, as required.

2. **Inversion:** For every inversion rule $r_1(x, y) \Rightarrow r_2(y, x)$, we combine the respective head and tail nodes of $r_1$ and $r_2$ such that $r_1^{(1)}$ and $r_2^{(2)}$, as well as $r_1^{(2)}$ and $r_2^{(1)}$, are each represented by one node. This makes that their corresponding boxes are equal, effectively capturing inversion patterns.

Following this step, $G$ now consists of at most $2|\mathbf{R}|$ nodes, and captures symmetry and inversion rules jointly. It now remains to define edges in $G$, as needed to later capture intersection and hierarchy rules. This is done analogously to the proof for generalized intersections (cf. Appendix C.1): First, the deductive closure of all intersection and hierarchy rules is computed, and the corresponding edges are encoded in $G$. Note that the resulting graph $G$ continues to capture inversion and symmetry, as these rules are encoded through nodes, and also encodes the deductive closure of all rules in $S_p$. Indeed, any box intersection imposed by the deductive closure of intersection and hierarchy rules with a node capturing a symmetry or inversion rule automatically implies a box intersection with the multiple boxes that the node represents. Hence, $G$ enables capturing symmetry and inversion rules a priori, as well as jointly sets up the necessary edges for hierarchy and inversion rules. Finally, we leverage the boxicity argument, and our final graph $G$, to obtain a box configuration where all the box intersections needed to later capture hierarchy and intersection rules are present (but not necessarily capturing hierarchy and intersection patterns at this stage), and which also successfully captures inversion and symmetry rules.

**Step 2: Anti-symmetry ($S_a$).** Anti-symmetry rules are captured by adding additional dimensions to the box configuration resulting from Step 1 to distinguish between the head and tail boxes of an anti-symmetric relation. $S$ is consistent, therefore only anti-symmetry rules not contradicting the set of rules $S_p \cup S_m$ can be given. For example, if symmetry rule $r(x, y) \Leftrightarrow r(y, x) \in S_p$, then $r(x, y) \Rightarrow \neg r(y, x) \notin S_a$. This is important, as it implies that no combination of hierarchy, inversion, intersection symmetry, and mutual exclusion rule can force an intersection between $r^{(1)}$ and $r^{(2)}$, for any anti-symmetric $r$, and thus, that subsequent steps in this proof preserve the anti-symmetry captured in this step.

We now capture anti-symmetry rules by dedicating a new "disjointness" dimension for all boxes, such that, for an anti-symmetric relation $r$, the box ranges for head and tail boxes are made disjoint in this dimension, i.e., $[l^{(1)}, u^{(1)}] \cap [l^{(2)}, u^{(2)}] = \phi$, and are set arbitrarily for all other relations, such that, for all rules in $S_p$, if an anti-symmetric $r$ is the head of a hierarchy rule $r_1 \implies r$, then the ranges of $r_1$ in this dimension respect the hierarchy and, for an intersection rule $r_1 \wedge r_2 \implies r$, then $r_1 \cap r_2 \subset r$. This initialization exists, as $S$ is consistent, so cannot create conflicting interval requirements for relations in rules. One can also observe this by considering this initialization a recursive pass through the rule sets affected by the anti-symmetric relations, where all other uninitialized relations in the deductive closure are not yet set. Hence, this new dimension captures $r^{(1)} \cap r^{(2)} = \phi$, so correctly captures anti-symmetry, and cannot be broken by subsequent rule-based box growth. It also is compatible with all symmetry and inversion rules, as box sharing is maintained. Hence, our current BoxE configuration captures any consistent set of anti-symmetry, symmetry, and inversion rules. Given $S_a$, at most $|S_a|$ additional dimensions are needed, and since at most $|\mathbf{R}|$ anti-symmetry rules can exist, the worst-case dimensionality of our configuration remains $O(|\mathbf{R}|\sqrt{\log(|\mathbf{R}|)})$. We now build on this result and show that the current configuration can be modified to additionally capture intersection and hierarchy rules.

**Step 3: Hierarchies and intersections.** Given the box configuration at the end of Step 2, we now apply the box reconfiguration procedure presented in the generalized intersections proof (cf. Appendix C.1) to capture all hierarchy and intersection rules in $S$. We also note that, since $S$ is consistent, no hierarchy and intersection rules force any inconsistency with the already captured symmetry, anti-symmetry and inversion rules, e.g., if $r_1(x,y) \Rightarrow r_1(y,x), r_2(x,y) \Rightarrow \neg r_2(y,x) \in S$, then $r_1(x,y) \Rightarrow r_2(x,y) \notin S$. Thus all symmetry, anti-symmetry, and inversion patterns, whose capture is based on structural concepts (box sharing and dedicated dimensions respectively), are preserved. In particular, box sharing is unaffected, and no box growth from this step can break the disjointness of anti-symmetric relation boxes, as $S$ is consistent. The completeness of the procedure with respect to hierarchy and intersection rules is also shown in Appendix C.1.

**Step 4: Mutual exclusion.** Given the BoxE configuration from Step 3, capturing rules from $S_p \cup S_a$, we also capture rules from $S_m$ with additional dimensions. Indeed, we show that this can be done using a BoxE configuration with $d = O(|\mathbf{R}|^2)$ dimensions. Starting from the configuration after the completion of Step 3, we now dedicate a single dimension per mutual exclusion rule, and capture this pattern as follows: For every mutual exclusion rule, we set a dimension, where $r_1$ and $r_2$ have disjoint range intervals $z_1, z_2 \subset [0,1]$, such that, without loss of generality, $z_1 = [z_{1,\min}, z_{1,\max}]$, $z_2 = [z_{2,\min}, z_{2,\max}]$ and $z_{2,\min} > z_{1,\max}$. Then, we set the range of every other box in the configuration at this new dimension analogously to Step 2 (i.e., arbitrarily, but in a rule-aware fashion) by repeating the box reconfiguration procedure in Step 3 for capturing hierarchy and intersection rules starting from the current configuration.

Note that anti-symmetry, symmetry, and inversion rules play no part in this step, as anti-symmetry rules are captured with dedicated dimensions as shown earlier, whereas symmetry and inversion rules are already enforced, and thus captured, through box sharing and equality.

Intuitively, this step first makes $r_1$ and $r_2$ mutually exclusive in one dimension, then recursively traverses the set of hierarchy and intersection rules, as in Step 3, to preserve the capturing of these rules in this new dimension specifically. Clearly, anti-symmetry remains true, since its dedicated dimension is not affected by the repetition of Step 3. Furthermore, since $S$ is consistent, all mutual exclusion rules in $S$ can be captured without causing inconsistency. In other words, rule sets such as $r_1(x,y) \Rightarrow r_2(x,y), r_1(x,y) \Rightarrow r_3(x,y)$, and $r_2(x,y) \wedge r_3(x,y) \Rightarrow \bot$ are not possible.

Hence, since $|S_m| \leq 0.5|\mathbf{R}|(|\mathbf{R}| - 1)$, the number of distinct pairs that can be selected from $\mathbf{R}$, a BoxE model with $d = 0.5|\mathbf{R}|(|\mathbf{R}| - 1) + |\mathbf{R}| + |\mathbf{R}|\sqrt{\log |\mathbf{R}|} = O(|\mathbf{R}|^2)$ dimensions can capture any consistent set of rules $S$ from the language of intersection, hierarchy, symmetry, anti-symmetry, mutual exclusion, and inversion rules.

We finally highlight one subtle, but important detail: Whereas the inference pattern language just described can be captured by a BoxE model having $d = O(|\mathbf{R}|^2)$ dimensions, some individual generalized patterns (inversion, hierarchy, symmetry, anti-symmetry, mutual exclusion) can be captured with even constant number of dimensions, and generalized intersection can be captured with $O(|\mathbf{R}|\sqrt{\log(|\mathbf{R}|)})$ dimensions. Hence, an interesting contrast in dimensionality requirements arises between capturing individual generalized inference patterns, capturing the language of Theorem 5.4, and capturing rule language of Theorem 5.3, which highlights the significantly larger requirements that capturing joint generalized requirements, and the potential existence of cycles, can impose on any embedding model.

# E Proof of Theorem 5.4 (Rule Injection)

We now prove that arbitrary sets $S_p$ of hierarchy, intersection, symmetry, and inversion rules can be injected into BoxE. To this end, we adapt the proof of Theorem 5.3 to this setting.

We start with a randomly initialized box configuration. First, we inject inversion and symmetry rules using box sharing: For symmetry rules, we set $r^{(1)} = r^{(2)}$, and for inversion rules, we set $r_1^{(1)} = r_2^{(2)}$ and $r_2^{(1)} = r_1^{(2)}$, and this can be done in linear time with respect to the number of inversion and symmetry rules. This achieves the same result as the node sharing in Step 1 of the proof of Theorem 5.3, except that the box configuration is a concrete random initialization, as opposed to an abstract configuration known to exist due to boxicity. We then proceed with the box reconfiguration procedure in Step 3 of this same proof to enforce hierarchy and intersection rules on top of inversion

Table 5: Properties of benchmark datasets FB15k-237, WN18RR, YAGO3-10, JF17K, and FB-AUTO.

| Dataset | $|\mathbf{E}|$ | $|\mathbf{R}|$ | Training Facts | Validation Facts | Testing Facts |
|---------|------|------|---------------|------------------|---------------|
| FB15k-237 | 14,541 | 237 | 272,115 | 17,535 | 20,466 |
| WN18RR | 40,943 | 11 | 86,835 | 3,034 | 3,034 |
| YAGO3-10 | 123,182 | 37 | 1,079,040 | 5,000 | 5,000 |
| JF17K | 29,257 | 327 | 61,911 | 15,822 | 24,915 |
| FB-AUTO | 3,388 | 8 | 6,778 | 2,255 | 2,180 |

and symmetry rules. This step is guaranteed to enforce these rules, and their deductive closure, as shown in Appendix D, and maintains box sharing, so preserves symmetry and inversion.

We now analyze the worst-case runtime complexity of the box reconfiguration procedure. We assume the worst-case, that any pairwise intersections should be expressible, and thus use a dimensionality $d = O(|\mathbf{R}|\sqrt{\log(|\mathbf{R}|)})$. The worst-case running time of the box reconfiguration procedure for enforcing a single hierarchy/intersection rule is $O(|\mathbf{R}|d) = O(|\mathbf{R}|^2\sqrt{\log(|\mathbf{R}|)})$, corresponding to the maximum number of boundary changes needed per call. However, this upper bound is independent of the number of rules in $S$, as no more than $O(|\mathbf{R}|d)$ steps can be made across all rules. Thus, the worst-case running time for rule injection across all hierarchy and intersection rules is $O(|\mathbf{R}|d) = O(|\mathbf{R}|^2\sqrt{\log(|\mathbf{R}|)})$.

Hence, rule injection for hierarchy and intersection rules runs at worst in near-quadratic time with respect to $|\mathbf{R}|$, a typically small number, irrespective of the number of these rules. This result, combined with the efficiency of enforcing symmetry and hierarchy, imply that BoxE can be efficiently injected with arbitrary sets of symmetry, inversion, hierarchy and intersection rules.

## F  Experimental Details

In this section, we give further details on the experiments that we have conducted. In particular, we report details of every dataset, the hyperparameter tuning setup used when training BoxE, as well as the final set of hyperparameters used in the configurations whose results we report in the paper. Finally, we report the complete set of results for KGC, higher-arity, and rule injection experiments, i.e., MR, MRR, Hits@1, Hits@3, and Hits@10. All reported results for the KGC and KBC experiments are average results from 3 training runs, and empirically have very small variance. In particular, all MRR values fluctuate by no more than 0.002 between runs across all datasets.

### F.1  Benchmark dataset details

In this subsection, we provide the details of of all benchmark datasets used in this paper (FB15k-237, WN18RR, YAGO3-10, JF17K, and FB-AUTO), namely the number of entities, relations, and facts in every split (training, validation, and test) in Table 5.

### F.2  Hyperparameter settings for BoxE experiments

BoxE is trained using the Adam optimizer [19], to optimize negative sampling loss [37]. Training for every run was conducted on a Haswell CPU node with 12 cores, 64 GB RAM, and a V100 GPU. Hyperparameter tuning was conducted over its learning rate $\lambda$, dimensionality $d$, loss margin $\gamma$, distance order $x$, and number of negative examples $m$. For all BoxE experiments, points and boxes were projected into the hypercube $[-1, 1]^d$, a bounded space, by simply applying the hyperbolic tangent function $\tanh$ element-wise on all final embedding representations.

Learning rate was varied between $10^{-6}$ and $10^{-2}$, with root values of 1,2,5 and exponents from -6 to -2, i.e., $10^{-6}, 2 \times 10^{-6}, 5 \times 10^{-6}$, etc. . Margin was varied between 3 and 24 inclusive, in increments of 1.5, and in increments of 1 between 3 and 6. Adversarial temperature was varied between the integer values of 1 and 4 inclusive, and the number of negative samples was varied between 50, 100, and 150. Across all knowledge graph datasets, we additionally ran experiments with *data augmentation*, such that, for every relation $r$, a distinct inverse relation $r'$ is defined, and every

Table 6: Hyperparameter settings of BoxE over different datasets.

| Dataset | Embedding Dimension | Margin | Learning Rate | Adversarial Temperature | Negative Samples | Distance Order | Batch Size | Data Augmentation |
|---|---|---|---|---|---|---|---|---|
| FB15k-237(u) | 500 | 12 | $1 \times 10^{-4}$ | 0.0 | 100 | 1 | 1024 | No |
| FB15k-237(a) | 1000 | 3 | $5 \times 10^{-5}$ | 4.0 | 100 | 2 | 1024 | No |
| WN18RR(u) | 500 | 5 | $1 \times 10^{-3}$ | 0.0 | 150 | 2 | 512 | No |
| WN18RR(a) | 500 | 3 | $1 \times 10^{-3}$ | 2.0 | 100 | 2 | 512 | No |
| YAGO3-10(u) | 200 | 10.5 | $1 \times 10^{-3}$ | 0.0 | 150 | 2 | 4096 | Yes |
| YAGO3-10(a) | 200 | 6 | $1 \times 10^{-3}$ | 2.0 | 150 | 2 | 4096 | Yes |
| JF17K(u) | 200 | 15 | $2 \times 10^{-3}$ | 0.0 | 100 | 2 | 1024 | N/A |
| JF17K(a) | 200 | 5 | $1 \times 10^{-4}$ | 2.0 | 100 | 2 | 1024 | N/A |
| FB-AUTO(u) | 200 | 18 | $2 \times 10^{-3}$ | 0.0 | 100 | 2 | 1024 | N/A |
| FB-AUTO(a) | 200 | 9 | $5 \times 10^{-4}$ | 2.0 | 100 | 2 | 1024 | N/A |
| SportsNELL | 200 | 6 | $1 \times 10^{-3}$ | 0.0 | 100 | 2 | 1024 | No |
| SportsNELL+RI | 200 | 6 | $1 \times 10^{-3}$ | 0.0 | 100 | 2 | 1024 | No |

Table 7: Complete KGC results for BoxE and competing models on FB15K-237 and WN18RR.

| Model | FB15K-237 | | | | | WN18RR | | | | |
|---|---|---|---|---|---|---|---|---|---|---|
| | MR | MRR | H@1 | H@3 | H@10 | MR | MRR | H@1 | H@3 | H@10 |
| TransE(u) [33] | - | .313 | - | - | .497 | - | .228 | - | - | .520 |
| RotatE(u) [37] | 185 | .297 | .205 | .328 | .480 | *3254* | *.470* | *.422* | *.488* | *.564* |
| BoxE(u) | **172** | **.318** | **.223** | **.351** | **.514** | **3117** | .442 | .398 | .461 | .523 |
| TransE(a) [37] | 170 | .332 | .233 | .372 | .531 | 3390 | .223 | .013 | .401 | .529 |
| RotatE(a) [37] | 177 | **.338** | **.241** | **.375** | .533 | 3340 | **.476** | **.428** | **.492** | **.571** |
| BoxE(a) | **163** | .337 | .238 | .374 | **.538** | 3207 | .451 | .400 | .472 | .541 |
| DistMult [33, 49] | - | .343 | - | - | .531 | - | .452 | - | - | .531 |
| ComplEx [33, 49] | - | .348 | - | - | .536 | - | **.475** | - | - | **.547** |
| TuckER [1] | - | **.358** | **.266** | **.394** | **.544** | - | .470 | **.443** | .482 | .526 |

fact $r(e_1, e_2)$ is augmented with another fact $r'(e_2, e_1)$. This setting, however, was only marginally beneficial on YAGO3-10, yielding a slightly improved MR.

Finally, the distance order was set to either 1 (Manhattan distance) or 2 (Euclidian distance), and batch sizes (for number of positive examples) were varied between all powers of two between $2^6$ and $2^{12}$ inclusive. Hyperparameters were initially selected randomly and tuned using grid search. The set of used hyperparameters in experiments is shown in Table 6.

Aside from the reported hyperparameter settings, we have also attempted to fix box sizes, either in a hard fashion or softly by setting maximum total size. Hard sizes were based on statistical popularity of relations, whereas soft totals were tuned. However, neither of these settings yielded any improvements, and in fact both have been mostly detrimental to performance. This, in fact, further highlights the importance of box size variability to obtaining good predictive performance. Interestingly, it also confirms that statistical popularity alone is not sufficient to establish optimal box sizing. We also remain very confident that BoxE performance can further improve in the future, as more dedicated empirical studies and more comprehensive and bespoke tuning methods are applied.

## F.3   Complete experimental results

The complete results for KGC experiments on FB15k-237, WN18RR, and YAGO3-10 are reported across Tables 7 and 8. Complete results for higher-arity KBC experiments on JF17K and FB-AUTO are reported in Table 9, and complete rule injection results for BoxE and BoxE+RI on the two SportsNELL evaluation sets are reported in Table 10.

Table 8: Complete KGC results for BoxE and competing models on YAGO3-10.

| Model | YAGO3-10 | | | | |
|---|---|---|---|---|---|
| | MR | MRR | H@1 | H@3 | H@10 |
| TransE(u) [33] | - | - | - | - | - |
| RotatE(u) [37] | *1116* | *.459* | *.360* | *.509* | *.651* |
| BoxE(u) | 1164 | **.567** | **.494** | **.611** | **.699** |
| TransE(a) [37] | - | - | - | - | - |
| RotatE(a) [37] | 1767 | .495 | .402 | .550 | .670 |
| BoxE(a) | **1022** | .560 | .484 | .608 | .691 |
| DistMult [33, 49] | 5926 | .34 | .24 | .38 | .54 |
| ComplEx [33, 49] | 6351 | .36 | .26 | .40 | .55 |
| TuckER [1] | *4423* | *.529* | *.451* | *.576* | *.670* |

Table 9: Complete KBC results on higher-arity datasets JF17K and FB-AUTO.

| Model | JF17K | | | | | FB-AUTO | | | | |
|---|---|---|---|---|---|---|---|---|---|---|
| | MR | MRR | H@1 | H@3 | H@10 | MR | MRR | H@1 | H@3 | H@10 |
| m-TransH | - | .446 | .357 | .495 | .614 | - | .728 | .727 | .728 | .728 |
| m-DistMult | - | .460 | .367 | .510 | .635 | - | .784 | .745 | .815 | .845 |
| m-CP | - | .392 | .303 | .441 | .560 | - | .752 | .704 | .785 | .837 |
| HypE | - | .492 | .409 | .533 | .650 | - | .804 | .774 | .823 | .856 |
| HSimplE | - | .472 | .375 | .523 | .649 | - | .798 | .766 | .821 | .855 |
| BoxE(u) | **363** | .553 | .467 | .596 | .711 | **110** | .837 | .804 | .858 | .895 |
| BoxE(a) | 372 | **.560** | **.472** | **.604** | **.722** | 122 | **.844** | **.814** | **.863** | **.898** |

Table 10: Complete rule injection experiment results on the SportsNELL full and filtered sets.

| Model | Full Set | | | | | Filtered Set | | | | |
|---|---|---|---|---|---|---|---|---|---|---|
| | MR | MRR | H@1 | H@3 | H@10 | MR | MRR | H@1 | H@3 | H@10 |
| BoxE | 17.4 | .577 | .478 | .623 | .780 | 19.1 | .713 | .661 | .732 | .824 |
| BoxE+RI | **1.74** | **.979** | **.968** | **.988** | **.997** | **5.11** | **.954** | **.938** | **.964** | **.984** |

# G    Additional Experimental Insights and Discussions

## G.1    Robustness experiment

In this experiment, we evaluate the dependence of BoxE on dimensionality $d$, to understand its prospective performance in a computationally restricted setting.

**Experimental setup.** We train BoxE with uniform negative sampling on YAGO3-10 using $d = \{25, 50, 100, 150, 200\}$. We only tune the margin and fix the learning rate, batch size, and number of negative samples to $10^{-3}$, 4096, and 150, respectively. $L2$ norm and data augmentation are used across all experiments. We report peak MRR recorded over the validation set. The final margins were $\gamma = 6$ for $d = 25$ and $\gamma = 10.5$ otherwise.

Figure 4: BoxE validation performance over YAGO3-10 versus dimensionality.

**Results.**    A plot of validation MRR versus dimensionality is drawn in Figure 4. BoxE maintains very strong performance, even at $d = 50$, rivaling that of state-of-the-art translational model RotatE, even

with just uniform negative sampling. Furthermore, it performs at near-optimal level with $d = 100$, and is already state-of-the-art on YAGO3-10 at this small dimensionality. Hence, BoxE proves to be very robust for performing knowledge base completion with restricted computational power.

## G.2 Box volume information for BoxE following training on YAGO3-10

In Table 11, we report the geometric mean of box volume across dimensions for all head and tail boxes for the 37 relations in YAGO3-10 following training. These numbers are computed from the same configuration whose results are reported in the main paper for BoxE(u). Note that, as explained in Appendix F, boxes are mapped to the space $[-1, 1]^d$ using the hyperbolic tangent function, so the geometric mean volume is upper-bounded by 2. From Table 11, we can make the following four very interesting observations:

First, we see that more popular relations, in terms of entities they connect, tend to be represented with larger boxes in the embedding space. This confirms our intuition that the boxes effectively define entity classes, and thus larger classes, are met with larger boxes in the embedding space. For example, the less popular relation hasWebsite has very small boxes of mean volume about 0.15, as it is only makes up 68 facts in the YAGO training dataset. By contrast, the relation created has both boxes with mean volume above 0.9, and appears in over 1,400 facts.

Second, we observe that the size of relation boxes also correlates with implicit entity types, in addition to relation popularity. Indeed, the relation playsFor, despite appearing over 300,000 times, only has box volumes 0.284 and 0.469 respectively, whereas isLeaderOf, with less than 1,000 facts, has a tail box of mean volume exceeding 1. This is due to the diversity in entity types appearing at these relations: For playsFor, head entities are athletes, which cluster together in a smaller region of the embedding space, and tail entities are football/sports clubs, which are more diverse, but still quite similar semantically. By contrast, head entities for isLeaderOf are individuals, with medium variability, but tail entities can be anything from very different countries (e.g., Mali, Kuwait) to cities, districts, and towns (e.g., Toronto, Oxnard (California)), to political parties and associations (e.g., Democratic Governors Association, Hungarian Communist Party), which are vastly different types of entities, and this results in an extremely large tail box for isLeaderOf.

Table 11: Geometric mean volume per dimension for all relation boxes in YAGO3-10 following training.

| Relation | Head Box | Tail Box |
|---|---|---|
| actedIn | 0.456 | 0.479 |
| created | 0.966 | 0.905 |
| dealsWith | 0.373 | 0.366 |
| diedIn | 0.383 | 0.480 |
| directed | 0.474 | 0.461 |
| edited | 0.461 | 0.441 |
| exports | 0.238 | 0.260 |
| graduatedFrom | 0.608 | 0.526 |
| happenedIn | 0.453 | 0.363 |
| hasAcademicAdvisor | 0.655 | 0.605 |
| hasCapital | 0.390 | 0.347 |
| hasChild | 0.299 | 0.761 |
| hasCurrency | 0.228 | 0.239 |
| hasGender | 0.669 | 0.688 |
| hasMusicalRole | 0.328 | 0.427 |
| hasNeighbor | 0.311 | 0.312 |
| hasOfficialLanguage | 0.213 | 0.255 |
| hasWebsite | 0.159 | 0.143 |
| hasWonPrize | 0.264 | 0.381 |
| imports | 0.249 | 0.241 |
| influences | 0.510 | 0.567 |
| isAffiliatedTo | 0.257 | 0.557 |
| isCitizenOf | 0.544 | 0.614 |
| isConnectedTo | 0.403 | 0.388 |
| isInterestedIn | 0.644 | 0.496 |
| isKnownFor | 0.632 | 0.623 |
| isLeaderOf | 0.446 | 1.005 |
| isLocatedIn | 0.496 | 0.547 |
| isMarriedTo | 0.923 | 0.924 |
| isPoliticianOf | 0.361 | 0.521 |
| livesIn | 0.536 | 0.341 |
| owns | 0.907 | 0.485 |
| participatedIn | 0.389 | 0.471 |
| playsFor | 0.284 | 0.469 |
| wasBornIn | 0.465 | 0.445 |
| worksAt | 0.498 | 0.488 |
| wroteMusicFor | 0.450 | 0.646 |

Third, we observe that relative box sizes accurately reflect the type of their underlying relation. More specifically, larger tail boxes tend to denote one-to-many relations, larger head boxes indicate a many-to-one relation, and similar sizes indicate many-to-many or one-to-one relations. This is especially evident for the one-to-many relations hasChild (0.299 vs 0.761), and isAffiliatedTo (0.257 vs 0.557), and for many-to-one relations isInterestedIn (0.644 vs 0.496), and graduatedFrom (0.608 vs 0.526).

Finally, we note that symmetric relations in YAGO3-10, namely hasNeighbor and isMarriedTo, are represented with near-identically sized boxes. This is a very important finding, as it indicates that

BoxE succesfully captures the symmetry inference pattern, for which a necessary condition is having identical head and tail boxes.

All in all, these results further highlight the interpretability of BoxE, in terms of capturing inference patterns, accurately inferring and portraying entity classes, and inferring and successfully modelling relation types, which other models are unable to achieve.

### G.3 Rule injection experiment

In this section, we provide additional information about the rule injection experiment presented in the paper. In particular, we give a more complete presentation of model convergence with and without rule injection, and provide further details on SportsNELL.

**Learning curves of BoxE, BoxE+RI.** The learning curves of BoxE, and Box+RI, defined with MRR as the performance metric, across the 2000 training epochs of the rule injection experiment, is shown in Figure 5. The two curves highlight a remarkable improvement stemming from injecting the SportsNELL ontology. Indeed, BoxE+RI converges to peak performance within 500 epochs, and mostly stablises its peak MRR following this point, whereas standard BoxE does not fully converge, even after the whole 2000 epochs have elapsed. Furthermore, the difference in performance between these two models is very significant. Hence, rule injection not only yields better-performing KBC systems, but also enables faster, more reliable training of these systems.

Figure 5: BoxE and BoxE+RI learning curves.

**Further details about SportsNELL.** SportsNELL initially consists of 181,936 facts, 11 relations and 4,252 sports-related entities, such that all its entities initially appear 50 or more times in NELL across these 11 relations. Its *logical closure* w.r.t the SportsNELL ontology is then computed., i.e., ontology rules are repeatedly applied to deduce new facts until no new facts can be deduced: new facts in the deductive closure are direct results of rule application, and thus their correct prediction indicates a good capturing of the underlying ontology. The resulting combined dataset, referred to as SportsNELL$^C$, contains a total of 326,650 facts.