[Reviews · NeurIPS 2020]

Review 1

Summary and Contributions: The paper uses previously developed box embeddings on the knowledge base / graph completion task. The paper explores which types of rules and relationships are expressible, capturable, and injectable in the box embedding model, finding this subset attractive. Experiments confirm that box embeddings outperform comparable approaches on a variety of tasks and provide a useful inductive bias when injecting new rules into a knowledge base. A box embedding is a representation relations as two boxes (or more for n-ary relations) in an embedding space, and points as two vectors (a basis point and a bump vector). For a relation r to apply to two entities the first entity's base point (translated by second entity's bump) has to be contained in the relation's first box and the second entity's base point (translated by the first entity's bump) has to be contained in the relation's second box. This representation, though a little hard to reason about, allows expressing a large set of relevant constraints on possible relations.

Strengths: The model is clearly very expressive (see table 5.1 for the types of patterns expressible in it) and seems relatively straightforward to train. The paper also very clearly specifies the distinctions between expressing a relation, capturing an inference pattern, and injecting a rule into a model (where some amount of nondeterminism and inductive bias is brought in). The experiments exploring the effects of these actions are illuminating.

Weaknesses: It's not easy to see how to compress these models. The proof for theorem 5.1 relies on an induction step adding dimensions for every rule you want to encode in the model. The definition of the box embedding can be a little hard to grok, and the base explanation is quite short. In writing this review I had to correct my paraphrase multiple times, and refer to the proofs for theorems 5.1 and 5.2 in the appendix to fully understand the implications. The main text would be improved with some examples of how the general constraints are encodable in terms of the boxes (like how symmetric relations have two identical boxes, etc).

Correctness: I read through the proofs in the appendix superficially and found no issues. The experimental results also show no problems.

Clarity: Yes, the paper is reasonably easy to read, though it took a few tries to really grok the definition of box relations as used here.

Relation to Prior Work: Yes all relevant prior work I am aware of has been cited.

Reproducibility: Yes

Additional Feedback:


Review 2

Summary and Contributions: This paper proposes to provide a novel link prediction model using box embedding. The proposed model seems to be immune to exiting shortcomings of KGE models such as theoretical inexpressivity, inability to properly capturing prominent inference patterns, and incapability in extending to higher-arity relations. The authors validate their proposed approach through several experiments. The contributions are as follows: 1) Providing a novel KGE model using box embedding. 2) Analyzing the performance of the proposed method in detailed experiments achieving several SOTA performances in link prediction task. 3) Studying rule injection and higher-arity relations in KGs, as extra features of the BoxE model.

Strengths: This paper reads well and the results appear sound. I personally find the capability of BoxE in learning higher-arity relations and rule injection very interesting. Furthermore, the provided approach outperforms previous methods in several benchmarks. Finally, the paper did a great job in thoroughly analyzing the proposed model theoretically and experimentally,

Weaknesses: The only drawbacks of this paper are novelty, considering the similarity of BoxE with previously introduced methods which use box embedding, and inadequate explanation on learning procedure of their model, such is loss function, and number of negative samples.

Correctness: The claims and methodology appear sound.

Clarity: I found the paper very well-written.

Relation to Prior Work: The authors discussed the connection to previous works clearly.

Reproducibility: Yes

Additional Feedback: My suggestions and questions are as follows: 1) It will be interesting to see the result of injecting rules on other KGs, such as WN18RR and FB15K-237. 2) I am curious about the authors' opinion on the interpretability of BoxE representations in comparison to other existing KGE models. 3) To see the capability of BoxE in capturing prominent inference patterns, it will be interesting to study the per-relation breakdown performance on different KGs.


Review 3

Summary and Contributions: Significant contribution to unifying relations of any arity for KBC, using per-entity displacements and modeling relations as box regions.

Strengths: + Rather creative box model for any-ary relations, unifying symmtry, asymmetry, anti-symmetry, and transitivity into single framework. + Thorough study of theoretical properties. + Comprehensive experiments and results.

Weaknesses: - Some minor notation malfunction, easily fixable. - Some material from appendix should be pushed into main paper.

Correctness: I did not check Thm 5.3 and 5.4 closely. The rest seem sound. Experimental methodology is standard.

Clarity: Very well written for the most part, minor fixes suggested.

Relation to Prior Work: Yes.

Reproducibility: Yes

Additional Feedback: Please number ALL equations for easy reference, at least in the preliminary submission. L139 Translational bumps are certainly very expressive, but a likely first reaction is that they are too expressive. Perhaps you need a couple sentences right here on how you control their power. L153 "for the sample KG, there are $4^2$ potential configurations" There are four entities and two binary relations. For each relation, each slot can be occupied by any one of four entities (assuming selectively reflexive and symmetric relations allowed). That is 16 possible worlds for each relation, so $16^2$ potential configurations --- is that wrong? L163 The notation can be improved and simplified. For relation $r$, you can use $\ell_r$ for the lower corner, $u_r$ for the upper corner, and $c_r$ for the center. In your notation, the center and width lose track of which relation it pertains to. Add notes on elementwise operations (particularly elementwise division and reciprocal) around equations between L165 and L166 to make them clearer. There is also no explanation for the form of the expression in the second line. The extreme case consistency mentioned in L166 is not enough. (The charts in the appendix makes the intention quite clear, but I could not `read' the equations as the charts on a rapid pass through the main paper. After seeing the charts, it seems like a 2-sided adaptation of ReLU variants that retain some guiding gradient.) The bump given to an entity by comrades in a tuple is itself independent of any parameterization of the relation. The relation parameters (center and `radius') encourage all bumped entities in a tuple to stay within its boundaries. In the proof of Theorem 5.1 as well as experiments, it seems you are depending on negative examples to control the extent of the boxes, rather than regularizing their widths or volumes. Are you convinced uniform negative sampling is best possible or most efficient?

[Author Response · NeurIPS 2020]

We thank the reviewers for their valuable and insightful feedback, and respond to their comments and questions below.

**Reviewer 1.** We are pleased that the different conceptual aspects of BoxE are clear, and that our experiments are illuminating in this regard. BoxE is indeed simple to train, only needing standard optimization (Adam) of its embeddings following loss computation, and is both fully expressive, and, quite uniquely, can jointly capture and inject a rich language of inference patterns, which is very desirable in practice for model interpretability and safety.

*Model expressivity and compression:* The bound in Theorem 5.1 is a worst-case bound that is only tight when all KB facts are fully independent, which is highly unlikely in practice. Indeed, real-world KB entities share several properties, and this dramatically reduces the required dimensionality. This is confirmed in our experiments, where all results use much smaller dimensionality than the bound (see Table 6). In fact, higher-arity experiments (see Section 6.2) are state-of-the-art with only 200 dimensions. Furthermore, we have evaluated model robustness in Appendix H.1, and observed that BoxE maintains strong performance even with just 50 dimensions on YAGO3-10. Thus, BoxE naturally compresses information within its embeddings, allowing it to perform well at lower dimensionality.

**Reviewer 2.** BoxE is indeed very different, as it handles arbitrary-arity KBs natively, and can capture and inject a rich class of logical rules. Please note that we present BoxE training details in Appendix G: We use cross-entropy loss, the Adam optimizer, and hyper-parameters (including negative samples) are in Table 6. We will mention this in the paper.

*Novelty of the model:* BoxE is substantially different from any existing box model. Box embedding models for entity classification cannot naturally scale beyond unary classification, and Query2Box yields a model like TransE on triples (and is primarily for querying rather than KBC). BoxE is novel in many ways: (i) it introduces translational bumps, without which existing box models are severely limited (see Section 4), (ii) it proposes a novel and unified way to represent multi-arity data, (iii) it has a powerful inductive capacity confirmed by a thorough theoretical analysis, and (iv) it allows for deductive inferences via rule injection, enabling a form of reasoning within gradient-based optimization. We will make these differences more explicit in the paper. Our response to reviewer suggestions is as follows:

*(1) Rule injection on other datasets:* We considered evaluating BoxE on other datasets, but unfortunately no rules/ontologies exist for standard KBC datasets. Hence, we evaluated rule injection on SportsNELL, a subset of NELL with a real-world ontology. We hope BoxE leads to the enriching of existing benchmarks with rule sets, so that future KBC works can be additionally evaluated on their ability to capture and/or inject rules.

*(2) Interpretability:* BoxE is a highly interpretable model, as a rich language of logical rules can be captured and read solely through box embeddings. Most importantly, inductive capacity (characterized logically) correlates highly with interpretability: the more rules a model can capture explicitly, the more interpretable it becomes. For example, we can simply read off hierarchies such as $r_1(x,y) \rightarrow r_2(x,y)$, as box subsumption in the space between $\mathbf{r_2}$ and $\mathbf{r_1}$. Box configurations can also inform us in various other ways, as we have empirically evaluated and observed in Appendix H.3: On YAGO3-10, BoxE captures symmetric relations through identical boxes, and its box sizes reflect many interesting relational properties (e.g., whether it is many-to-one, one-to-many, etc.). Overall, BoxE is significantly more interpretable than existing KBC models, but there is need for more interpretability also for box embeddings. We will explicitly mention the connection between inductive capacity and interpretability in the paper.

*(3) Per-relation break-down:* In Appendix H.3, BoxE learns identical boxes for symmetric relations HASNEIGHBOR and ISMARRIED. We have similar observations for symmetric relations in WN18RR. We will highlight these findings in the paper. Generally, our findings support the need for more systematic evaluations against inference patterns.

**Reviewer 4.** BoxE is indeed a strong unifying model for KBC that generalizes to arbitrary KBs and formalizes the study of inductive capacity. We will rectify all typing (equation numbering and element-wise operators) and notation (box centers and corners) malfunctions, and will transfer figures for loss and base model explanations from the appendix.

*Number of possible configurations:* Yes, "$4^2$ possible configurations" is a typing error, which should say "$4^2$ possible facts". Each fact can be true or false, so there are $2^{4^2}$ possible worlds/configurations (i.e., all subsets). We fixed this.

*Power of bumps:* Entity bumps are indeed powerful, but they are unique per entity and relation-independent, which enforces structure sharing and restricts their power implicitly, yielding a strong generalization. We also experimented with relation-specific bumps, which led to overfitting, suggesting that they were "too powerful" for the standard datasets.

*Negative examples and regularization:* Negative examples are indeed what is relied on to maintain reasonable box sizes. We have investigated both uniform and self-adversarial sampling, with the latter empirically delivering better performance in most cases. Therefore, we believe that other sampling techniques could indeed deliver better performance than uniform sampling. In terms of regularization, we have trained BoxE with fixed-size boxes (see Appendix G.2), and have also deployed regularizers on box size and position, but these approaches have not delivered any empirical gain relative to the presented setup. Nonetheless, we are confident that BoxE can further be improved with more sophisticated training and tuning techniques, which is a very interesting area for future work.

[Meta-Review · NeurIPS 2020]

The paper aims to improve knowledge base modelling. In this regards, authors propose a rather ingenious use of box embeddings as the latent representation for the relations. Specifically, each n-ary relation is represented by n boxes and each entity is represented by two vectors. Having a pair of vectors is very powerful, as they allow us to model complex interactions across entities. In particular authors show how their proposed box embeddings can simultaneously handle symmetry, asymmetry, anti-symmetry, and transitivity. No previous framework is claimed to be as flexible nor capable of handling all these patterns. The reviewers also liked the thorough study of theoretical properties. Paper also suggests ways to incorporate prior knowledge into the proposed box embeddings. Finally experiments for link prediction and rule injections were carried out on standard KB datasets. On high degree relations proposed method improves state-of-the-art results significantly as expected from design and injecting prior knowledge as rules also seem to help significantly. Overall reviewers reached a consensus to accept the paper. For the camera ready version, please tune baselines more for Yago and also mention that results are your evaluations and not as reported in original paper (as is the case for other datasets). For example I could get better result for Yago using RotatE.